# FedDAR: Federated Domain-Aware Representation Learning

**Aoxiao Zhong[1][3]***, **Hao He[2]***, **Zhaolin Ren[1], Na Li[1], and Quanzheng Li[3]**

[1]Harvard University     [2]Massachusetts Institute of Technology
[3]Massachusetts General Hospital/Harvard Medical School

`aoxiaozhong@g.harvard.edu, haohe@mit.edu, zhaolinren@g.harvard.edu`
`nali@seas.harvard.edu, li.quanzheng@mgh.harvard.edu`

## ABSTRACT

Cross-silo Federated learning (FL) has become a promising tool in machine learning applications for healthcare. It allows hospitals/institutions to train models with sufficient data while the data is kept private. To make sure the FL model is robust when facing heterogeneous data among FL clients, most efforts focus on personalizing models for clients. However, the latent relationships between clients' data are ignored. In this work, we focus on a special non-iid FL problem, called *Domain-mixed FL*, where each client's data distribution is assumed to be a mixture of several predefined domains. Recognizing the diversity of domains and the similarity within domains, we propose a novel method, FedDAR, which learns a domain shared representation and domain-wise personalized prediction heads in a decoupled manner. For simplified linear regression settings, we have theoretically proved that FedDAR enjoys a linear convergence rate. For general settings, we have performed intensive empirical studies on both synthetic and real-world medical datasets which demonstrate its superiority over prior FL methods. Our code is available at `https://github.com/zlz0414/FedDAR`.

## 1 INTRODUCTION

Federated learning (FL) (McMahan et al., 2017a) is a machine learning approach that allows many clients(e.g. mobile devices or organizations) to collaboratively train a model without sharing the data. It has great potential to resolve the dilemma in real-world machine learning applications, especially in the domain of healthcare. A robust and generalizable model in medical application usually requires a large amount of diverse data to train. However, collecting a large-scale centralized dataset could be expensive or even impractical due to the constraints from regulatory, ethical and legal challenges, data privacy and protection (Rieke et al., 2020).

While promising, applying FL to real-world problems has many technical challenges. One eminent challenge is data heterogeneity. Data across the clients are assumed to be independently and identically distributed (iid) by many FL algorithms. But this assumption rarely holds in the real world. It has been shown that non-iid data distributions will cause the failure of standard FL strategies such as FedAvg (Jiang et al., 2019; Sattler et al., 2020; Kairouz et al., 2019; Li et al., 2020). As an ideal model that can perform well on *all* clients may not exist, it requires FL algorithms to personalize the model for different data distributions.

Prior theoretical work (Marfoq et al., 2021) shows that it is impossible to improve performances on all clients without making assumptions about the client's data distributions. Past works on personalized FL methods (Marfoq et al., 2021; Sattler et al., 2020; Ghosh et al., 2020; Mansour et al., 2020; Deng et al., 2020) make their own assumptions and tailor their methods to those assumptions. In this paper, we propose a new and more realistic assumption where each client's data distribution is a mixture of several predefined domains. We call our problem setting *Domain-mixed FL*. It is inspired by the fact that the diversity of the medical data can be attributed to some known concept of domains, e.g., different demographic/ethnic groups of patients (Szczepura, 2005; Ranganathan & Bhopal, 2006; NHS, 2004), different manufacturers or protocols/workflows of image scanners (Mårtensson et al.,

---

*Equal contribution

2020; Ciompi et al., 2017) and so on. It is necessary to address the ubiquitous issue of domain shifts among ethic groups (Szczepura, 2005; Ranganathan & Bhopal, 2006; NHS, 2004) or vendors (Yan et al., 2019; Garrucho et al., 2022; Guan & Liu, 2021) in healthcare data. Despite of the domain shifts, same domain at different clients are usually considered to have the same distribution. The data heterogeneity between FL clients actually comes from the distinct mixtures of diverse domains at clients. These factors motivate us to *personalize model for each domain instead of client.*

Although our method is inspired by healthcare applications where the domain shifts issue is well-known and domain labels are very basic and accessible, we believe that it can be generally applied to other domains like finance or recommendation systems where users/humans with different demography are involved (Ding et al., 2021; Asuncion & Newman, 2007). However it would require a deep understanding of the data and background knowledge to verify the data distribution assumption as well as the accessibility of the domain label.

FedEM(Marfoq et al., 2021) and FedMinMax(Papadaki et al., 2021) makes similar assumption on data distribution as ours. However, FedEM assumes the domains are unknown and tries to learn a linear combination of several shared component models with personalized mixture weights through an EM-like algorithm. FedMinMax doesn't acknowledge the domain shift between domains and still aims to learn one shared model across domains by adapting minmax optimization to FL setting .

**Our Contributions.** We formulate the proposed problem setting, *Domain-mixed FL*. Through our analysis, we find prior FL methods, both generic FL methods like FedAvg (McMahan et al., 2017a), and personalized FL methods like FedRep (Collins et al., 2021), are sub-optimal under our setting. To address this issue, we propose a new algorithm, *Federated Domain-Aware Representation Learning (FedDAR)*. FedDAR learns a shared model for all the clients but embedded with domain-wise personalized modules. The model contains two parts: an shared encoder across all domains and a multi-headed predictor whose heads are associated with domains. For an input from one specific domain, the model extracts representation via the shared encoder and then use the corresponding head to make the prediction. FedDAR decouples the learning of the encoder and heads by alternating between the updates of the encoder and the heads. It allows the clients to run many local updates on the heads without overfitting on domains with limited data samples. This also leads to faster convergence and better performed model. FedDAR also adapts different aggregation strategies for the two parts. We use a weighted average operation to aggregate the local updates for the encoder. With additional sample re-weighting, the overall training objective is equally weighted for each domain to encourage the fairness among domains. While for the heads, we propose a novel second-order aggregation algorithm to improve the optimality of aggregated heads.

We theoretically show our method enjoys nice properties like linear convergence and small sample complexity in a linear case. Through extensive experiments on both synthetic and real-world datasets, we demonstrate that FedDAR significantly improves performance over the state-of-the-art personalized FL methods. To the best of our knowledge, our paper is among the first efforts in domain-wise personalized federated learning that achieve such superior performance.

## 2 RELATED WORK

Besides the literature we have discussed above, other works on personalization and fairness in federated learning are also closely related to our work.

**Personalized Federated Learning.** Personalized federated learning has been studied from a variety of perspectives: i) local fine-tuning (Wang et al., 2019; Yu et al., 2020) ii) meta-learning (Chen et al., 2018; Fallah et al., 2020; Jiang et al., 2019; Khodak et al., 2019) iii) local/global model interpolation (Deng et al., 2020; Corinzia et al., 2019; Mansour et al., 2020). iv) clustered FL that partition clients into clusters and learn optimal model for each cluster (Sattler et al., 2020; Mansour et al., 2020; Ghosh et al., 2020)(Zhu et al., 2021). v) Multi-Task Learning(MTL) (Vanhaesebrouck et al., 2017; Smith et al., 2017; Zantedeschi et al., 2020) (Hanzely & Richtárik, 2020; Hanzely et al., 2020; T Dinh et al., 2020; Huang et al., 2021; Li et al., 2021a) vi) local representations or heads for clients (Arivazhagan et al., 2019; Liang et al., 2020; Collins et al., 2021)(Luo et al., 2022). vii) personalized model through hypernetwork or super model (Shamsian et al., 2021; Chen & Chao, 2021; Xu et al., 2022). The personalization module in our approach is similar to vi) and (Zhu et al., 2021) with a multi-branch network. However, the targets we are personalizing the model for are the domains instead of clients.

**Fairness in Federated Learning.** There are two commonly used definitions of fairness in existing FL works. One is client fairness, usually formulated as *client parity (CP)*, which requires clients to have similar performance. A few works (Li et al., 2021a; 2019; Mohri et al., 2019; Yue et al., 2021; Zhang et al., 2020) have studied on this. Another is group fairness. In the centralized setting, the fundamental tradeoff between group fairness and accuracy has been studied (Menon & Williamson, 2018; Wick et al., 2019; Zhao & Gordon, 2019), and various fair training algorithms have been proposed(Roh et al., 2020; Jiang & Nachum, 2020; Zafar et al., 2017; Zemel et al., 2013; Hardt et al., 2016). Since the notions of group fairness is the same in FL setting, most of existing FL works adapt methods from centralized setting (Zeng et al., 2021; Du et al., 2021; Gálvez et al., 2021; Chu et al., 2021; Cui et al., 2021). In this work, our method is not designed specifically for certain group fairness notions like demographic parity. Instead, we aim to achieve the best possible performance for each domain through personalization, admitting the difference between data domains. Moreover, our concept of data domains is not limited as demographic groups. It can also be applied to any other mixture of domain data, as long as our assumptions hold.

## 3 PROBLEM: DOMAIN-MIXED FEDERATED LEARNING

**Notations.** Federated learning involves multiple clients. We denote number of clients as $n$. We use $i \in [n] \triangleq \{1, 2, ..., n\}$ to index each client. Client $i$ has a local data distribution $\mathcal{D}_i$ which induces a local learning objective, i.e., the expected risk $\mathcal{R}_i(f) = \mathbb{E}_{(\boldsymbol{x}_i, y_i) \sim \mathcal{D}_i}[\ell(f(\boldsymbol{x}_i), y_i)]$, where $f : \mathcal{X} \to \mathcal{Y}$ is the model mapping the input $\boldsymbol{x} \in \mathcal{X}$ to the predicted label $f(\boldsymbol{x}) \in \mathcal{Y}$ and $\ell : \mathcal{Y} \times \mathcal{Y} \to \mathbb{R}$ is a generic loss function. In real practice, client $i \in [n]$ has a finite number, say $L_i$, of data samples, i.e., $\mathcal{S}_i = \{(\boldsymbol{x}_i^j, y_i^j)\}_{j=1}^{L_i}$. $L = \sum_{i=1}^{n} L_i$ denotes the total number of data samples.

**Problem Formulation of Domain-mixed Federated Learning.** We introduce a new formulation of FL problem by assuming each clients' local data distribution is a weighted mixture of $M$ domain specific distributions. Specifically, we use $\{\tilde{\mathcal{D}}_m\}_{m=1}^{M}$ to denote data distributions from $M$ predefined domains. For client $i$, its local data distribution is $\mathcal{D}_i = \sum_m \pi_{i,m} \tilde{\mathcal{D}}_m$ where the mixing coefficients $\pi_{i,m}$ stand for the probabilities of client $i$'s data sample coming from domain $m$. Take medical application as an example, different hospitals are clients and different ethnic groups are domains. Each ethnic group have different health data while each hospital's data is a mix of ethnic group data.

Further, the domains of the data samples are assumed to be known. We use a triplet of variables $(\boldsymbol{x}, y, z)$ to represent the input features, label and domain. The goal of our problem is to learn a model $f(\boldsymbol{x}, z)$ that can perform well in every domain, as shown by the following learning objective,

$$\min_f \mathcal{R}(f) := \frac{1}{M} \sum_{m=1}^{M} \mathcal{R}_m(f(\cdot, m)) \tag{1}$$

where $\mathcal{R}_m(f(\cdot, m)) = \mathbb{E}_{(\boldsymbol{x}, m) \sim \tilde{\mathcal{D}}_m}[\ell(f(\boldsymbol{x}, m), y)]$. Our problem focuses on the setting that each domain have a different conditioned label distribution, i.e., $P_m(y|\boldsymbol{x})$ is different in each domain $m$.

### 3.1 COMPARISON WITH PRIOR DOMAIN-UNAWARE FL PROBLEM FORMULATIONS

Our FL problem introduces the concept of the domain and focuses on the model's performance in each domain. Many prior FL formulations does not recognize the existence of the domains. For example, the original federated learning algorithms like FedAvg (McMahan et al., 2017a), FedProx (Li et al., 2020) learn a globally shared model that via minimizing the averaged risk, i.e., $\min_f \frac{1}{n} \sum_i \mathcal{R}_i(f)$. Some variants consider the fairness across the clients. To do so they optimize the worst client's performance, instead of the averaged performance, i.e., $\min_f \max_i \mathcal{R}_i(f)$. Further, personalized FL algorithms, such as FedRep (Collins et al., 2021), customize the model's prediction for each client whose objective is $\min_{f_i : i \in [n]} \frac{1}{n} \sum_{i=1}^{n} \mathcal{R}_i(f_i)$.

All the FL algorithms mentioned above will lead sub-optimal solutions to our problem since they do not make *domain specific* predictions. We illustrate this point by the following toy example of linear regression: We assume the data in $m$'th domain is generated via the following procedure: $\boldsymbol{x} \in \mathbb{R}^d$ is i.i.d sampled from a distribution $p(\boldsymbol{x})$ with mean zero and covariance $\boldsymbol{I}_d$. The label $y \in \mathbb{R}$ obeys $y = \boldsymbol{x}^\top \boldsymbol{B}^* \boldsymbol{w}_m^*$ where $\boldsymbol{B}^* \in \mathbb{R}^{d \times k}$ is ground truth linear embedding shared by all domains, and $\boldsymbol{w}_m^* \in \mathbb{R}^k$ is the linear head specific to domain $m$. Under this setting, $\tilde{\mathcal{D}}_m$ stands for data $(\boldsymbol{x}, y)$

---

**Algorithm 1** FEDDAR

---

**Input:** Data $\mathcal{S}_{1:n}$; number of local updates $\tau_h$ for the heads, $\tau_\phi$ for representation; number of communication rounds $T$; learning rate $\eta$.

Initialize representation and heads $\phi^0, h_1^0, ..., h_M^0$.

**for** $t = 1, 2, ..., T$ **do**

    Server sends $\phi^{t-1}, h_1^{t-1}, ..., h_M^{t-1}$ to the $n$ clients;

    **for** client $i = 1, 2, ..., n$ **in parallel do**

        Client $i$ initializes $h_{i,m}^{t,0} \leftarrow h_m^{t-1}, \forall m \in [M]$.

        **for** $s = 1$ **to** $\tau_h$ **do**

            $h_{i,m}^{t,s} \leftarrow \text{GRD}(\hat{\mathcal{R}}_{i,m}(h_{i,m}^{t,s-1} \circ \phi^{t-1}), h_{i,m}^{t,s-1}, \eta)$, for all $m \in [M]$.

        **end for**

        Client $i$ sends updated heads $h_{i,m}^{t,\tau_h}$ and Hessians $\boldsymbol{H}_{\mathcal{R}_{i,m}}(h_{i,m}^{t,\tau_h})$ to the server.

    **end for**

    Server aggregate the heads for each domain:

    **for** $m \in [M]$ **do**

        $h_m^t \leftarrow \text{HEADAGG}(\{h_{1,m}^{t,\tau_h}, \boldsymbol{H}_{\mathcal{R}_{1,m}}(h_{1,m}^{t,\tau_h})\}_{i=1}^n)$ via Equation 8.

    **end for**

    Server sends $h_1^t, ..., h_M^t$ to the $n$ clients;

    **for** client $i = 1, 2, ..., n$ **in parallel do**

        **for** $s = 1$ **to** $\tau_\phi$ **do**

            $\phi_i^{t,s} \leftarrow \text{GRD}(\hat{\mathcal{R}}_i(\phi_i^{t,s-1}, \{h_m^t\}_{m=1}^M), \phi_i^{t,s-1}, \eta)$.

        **end for**

        Client $i$ sends updated representation $\phi_i^t = \phi_i^{t,\tau_\phi}$ to server.

    **end for**

    Server computes the new representation via averaging $\phi^t \leftarrow \sum_{i=1}^n \frac{L_i}{L} \times \phi_i^t$.

**end for**

---

where $x \sim p(\boldsymbol{x})$ and $y = \boldsymbol{x}^\top \boldsymbol{B}^* \boldsymbol{w}_m^*$. For each client, the local data $\mathcal{D}_i$ is a mix of data from different domains with mixed coefficients, i.e., $\mathcal{D}_i = \sum_m \pi_{i,m} \tilde{D}_m$.

**FedAvg:** learns a single model $\boldsymbol{B}$ and $\boldsymbol{w}$ across the all clients via the following objective,

$$\min_{\boldsymbol{B},\boldsymbol{w}} \frac{1}{2n} \sum_{i \in [n]} \mathbb{E}_{(\boldsymbol{x},y) \sim \mathcal{D}_i}(y - \boldsymbol{x}^\top \boldsymbol{B}\boldsymbol{w})^2 = \sum_{i \in [n], m \in [M]} \frac{\pi_{i,m}}{2n} \mathbb{E}_{(\boldsymbol{x},y) \sim \tilde{D}_m}(y - \boldsymbol{x}^\top \boldsymbol{B}\boldsymbol{w})^2 \quad (2)$$

**FedRep:** learns shared representation $\boldsymbol{B}$ and separated heads $\boldsymbol{w}_i$ for each clients $i$ rather than for each domain $m$,

$$\min_{\boldsymbol{B},\boldsymbol{w}_1,...,\boldsymbol{w}_n} \frac{1}{2n} \sum_{i \in [n]} \mathbb{E}_{(\boldsymbol{x},y) \sim \mathcal{D}_i}(y - \boldsymbol{x}^\top \boldsymbol{B}\boldsymbol{w}_i)^2 = \sum_{i \in [n], m \in [M]} \frac{\pi_{i,m}}{2n} \mathbb{E}_{(\boldsymbol{x},y) \sim \tilde{\mathcal{D}}_m}(y - \boldsymbol{x}^\top \boldsymbol{B}\boldsymbol{w}_i)^2 \quad (3)$$

**FedDAR:** In contrast, in the linear case, our proposed method, FedDAR, which will be introduced next, learns a shared representation $\boldsymbol{B}$ and separate heads $\boldsymbol{w}_m$ for each domain $m$,

$$\min_{\boldsymbol{B},\boldsymbol{w}_1,\cdots,\boldsymbol{w}_m} \frac{1}{2M} \sum_{i \in [n]} \sum_{m \in [M]} \frac{\pi_{i,m}}{\sum_{i'} \pi_{i',m}} \mathbb{E}_{(\boldsymbol{x},y) \sim \tilde{\mathcal{D}}_m}(y - \boldsymbol{x}^\top \boldsymbol{B}\boldsymbol{w}_m)^2 \quad (4)$$

From the above formulations, we can see that FedAvd and FedRep are not able to achieve the zero error in our domain-mixed FL problem.

## 4 PROPOSED METHOD: FEDDAR

To solve the Domain-mixed FL problem, we propose a new method called, *Federated Domain-Aware Representation Learning* (FedDAR). In the following, we first introduce the model, learning objective and the details of the federated optimization algorithm.

## 4.1 Algorithm Overview

Our model is made of a shared encoder $\phi(\cdot; \boldsymbol{\theta})$ and $M$ domain specific heads $h_m(\cdot; \boldsymbol{w}_m)$ whose are parameterized by neural networks with the weights $\boldsymbol{\theta}$ and $\boldsymbol{w}_m, \forall m \in [M]$. According to our problem formation in Equation 1, our algorithm aims to solve the following optimization,

$$\min_{\phi, h_1, \ldots, h_M} \mathcal{R}(\phi, h_1, \ldots, h_M) := \frac{1}{M} \sum_{m=1}^{M} \mathcal{R}_m(h_m \circ \phi) \tag{5}$$

We decouple the training between encoder and heads. Specifically, we alternates the learning between the encoder and the heads. The learning is done federatedly and has two conventional steps: (1) local updates; (2) aggregation at the server. Algorithm 1 shows the pseudocode code.

**Empirical Objectives with Re-weighting.** Empirically, the objectives are estimated via the finite data samples at each client. We use $\mathcal{S}_{i,m}$ to denote the set of samples from domain $m$ in client $i$, with $L_{i,m} := |\mathcal{S}_{i,m}|$ denoting the sample size. Further, $L_i := \sum_{m=1}^{M} L_{i,m}$ is the number of samples in client $i$ while $L_m := \sum_{i=1}^{n} L_{i,m}$ is the total number of samples belonging to domain $m$ across all the clients. We denote the empirical risk at client $i$ specific to domain $m$ as $\hat{\mathcal{R}}_{i,m}(h_m \circ \phi) := \frac{1}{L_{i,m}} \sum_{(\boldsymbol{x},y) \in \mathcal{S}_{i,m}} \ell(h_m \circ \phi(\boldsymbol{x}), y)$. The empirical risk at client $i$ is designed as $\hat{\mathcal{R}}_i(\phi, h_1, \ldots, h_M) = \sum_m \frac{L_{i,m}}{L_i} u_m \hat{\mathcal{R}}_{i,m}(h_m \circ \phi)$, where $u_m = \frac{L}{L_m M}$ re-weights the risk for each domain. Combining commonly used weighted average FL objective $\hat{\mathcal{R}}(\phi, h_1, \ldots, h_M) = \sum_{i=1}^{n} \frac{L_i}{L} \hat{\mathcal{R}}_i(\phi, h_1, \ldots, h_M)$, the overall empirical risk is derived as the following,

$$\hat{\mathcal{R}}(\phi, h_1, \ldots, h_M) := \sum_{i=1}^{n} \frac{L_i}{L} \hat{\mathcal{R}}_i(\phi, h_1, \ldots, h_M) = \frac{1}{M} \sum_{m=1}^{M} \hat{\mathcal{R}}_m(h_m \circ \phi), \tag{6}$$

where $\hat{\mathcal{R}}_m(h_m \circ \phi) := \sum_{i=1}^{n} \frac{L_{i,m}}{L_m} \hat{\mathcal{R}}_{i,m}(h_m \circ \phi)$. This is consistent with Equation 5.

## 4.2 Local Updates at Clients

In each communication round, clients use gradient descent methods to optimize representation $\phi(\cdot; \boldsymbol{\theta})$ and local heads $h_m(\cdot; \boldsymbol{w}_m)$ for $m \in [M]$ alternately. We use $t$ to denote the current round. For a module $f$, $f^{t-1}$ denotes its optimized version after $t-1$ rounds. Each round has multiple gradient descent iterations. We use $f^{t,s}$ to denote the module in round $t$ after $s$ iterations. Since the updates are made locally, clients maintain their own copies of both modules, we use subscripts $i$ to index local copy at client $i$, e.g., $f_i^{t,s}$. We use GRD to denote a generic gradient-base optimization step which takes three inputs: *objective function*, *variables*, *learning rate* and maps them into a new module with updated variables. For example, the vanilla gradient descent has the form $\text{GRD}(\mathcal{L}(f_{\boldsymbol{w}}), f_{\boldsymbol{w}}, \eta) = f_{\boldsymbol{w} - \eta \nabla_{\boldsymbol{w}} \mathcal{L}(f_{\boldsymbol{w}})}$.

**For the heads**, client $i$ performs $\tau_h$ local gradient-based updates to obtain optimal head given the current shared encoder $\phi^{t-1}$. For $s \in [\tau_h]$, client $i$ updates via $h_{i,m}^{t,s} \leftarrow \text{GRD}(\hat{\mathcal{R}}_{i,m}(h_{i,m}^{t,s-1} \circ \phi^{t-1}), h_{i,m}^{t,s-1}, \eta)$. **For the shared encoder**, the clients executes $\tau_\phi$ local updates. Specifically, for $s \in [\tau_\phi]$, client $i$ updates the local copy of the encoder via $\phi_i^{t,s} \leftarrow \text{GRD}(\hat{\mathcal{R}}_i(\phi_i^{t,s-1}, \{h_m^t\}_{m=1}^M), \phi_i^{t,s-1}, \eta)$. The re-weighting mentioned in last section is implemented by re-weighting each sample with $u_m$ when calculating the loss function.

## 4.3 Aggregation at Server

We introduce two strategies: (1) weighted average (WA); (2) second-order aggregation (SA).

**Weighted average** means the aggregated model parameters are the average of the local model's parameters weighted by the number of data samples. Specifically, for the shared encoder, we have $\boldsymbol{\theta}^t = \sum_{i=1}^{n} \frac{L_i}{L} \boldsymbol{\theta}^{t-1}$. Similarly for each head, we have $\boldsymbol{w}_m^t = \sum_{i=1}^{n} \frac{L_{i,m}}{L_m} \boldsymbol{w}_{m,i}^{t-1}$.

**Second-order aggregation** is a more complex strategy. Ideally, we want the head aggregation generates the globally optimal model given a set of locally optimal model, as shown in the following,

$$\boldsymbol{w}^* \in \arg\min_{\boldsymbol{w}} \mathcal{J}(\boldsymbol{w}) \triangleq \sum_{i=1}^{n} \alpha_i \mathcal{J}_i(\boldsymbol{w}), \quad \text{given } \boldsymbol{w}_i^* = \arg\min_{\boldsymbol{w}} \mathcal{R}_i(\boldsymbol{w}) \ \forall i \in [n]. \tag{7}$$

where $\mathcal{J}_i$ is $i$'th client's virtual objective, $\alpha_i := L_i/L$ is the importance of the client, $L_i$ is the number of data samples. We call $\mathcal{J}_i$ the virtual objective to distinguish it from the real learning objective $\mathcal{R}_i$. The virtual objective is defined as an objective that the local updates give the optimal solution w.r.t it. It is introduced since the local updates during two aggregated are not guaranteed to optimize the head to optimal w.r.t the real objective. For example, if each local update is single step gradient descent with a learning rate $\eta$, i.e., $\boldsymbol{w}_i^{t+1} = \boldsymbol{w}^t - \eta\nabla_{\boldsymbol{w}}\mathcal{R}_i(\boldsymbol{w}^t)$. Then the virtual objective becomes $\mathcal{J}_i(\boldsymbol{w}) = \mathcal{R}_i(\boldsymbol{w}^t) + (\boldsymbol{w}-\boldsymbol{w}^t)^\top\nabla_{\boldsymbol{w}}\mathcal{R}_i(\boldsymbol{w}^t) + \frac{1}{2\eta}\|\boldsymbol{w}-\boldsymbol{w}^t\|_2^2$ which satisfies $\boldsymbol{w}_i^{t+1} \in \arg\min_{\boldsymbol{w}}\mathcal{J}_i(\boldsymbol{w})$. Such a virtual objective leads the solution of problem 7 to $\boldsymbol{w}^* = \frac{1}{n}\sum_{i=1}^n \boldsymbol{w}_i^*$ which is the simple averaging strategy.

However, in real practice, the local updates are usually more complicated which makes the virtual objective closer to the true objective. We consider the case that the virtual objective is the second-order Taylor expansion of the true objective, i.e., $\mathcal{J}(\boldsymbol{w}) = \mathcal{R}(\boldsymbol{w}_t) + (\boldsymbol{w} - \boldsymbol{w}^t)^\top\nabla_{\boldsymbol{w}}\mathcal{R}(\boldsymbol{w}^t) + \frac{1}{2}(\boldsymbol{w} - \boldsymbol{w}^t)^\top\boldsymbol{H}_{\mathcal{R}}(\boldsymbol{w}^t)(\boldsymbol{w} - \boldsymbol{w}^t)$ where $\boldsymbol{H}_{\mathcal{R}}$ is the Hessian matrix. Then each round of local update equivalent to a Newton-like step, $\boldsymbol{w}_i^{t+1} = \boldsymbol{w}^t - \boldsymbol{H}_{\mathcal{R}_i}(\boldsymbol{w}^t)^{-1}\nabla_{\boldsymbol{w}}\mathcal{R}_i(\boldsymbol{w}^t)$. While $\boldsymbol{w}^{t+1} = \boldsymbol{w}^t - \boldsymbol{H}_{\mathcal{R}}(\boldsymbol{w}^t)^{-1}\nabla_{\boldsymbol{w}}\mathcal{R}(\boldsymbol{w}^t)$ is the desired globally optima. Leveraging the fact that, $\nabla_{\boldsymbol{w}}\mathcal{R}(\boldsymbol{w}) = \sum_{i\in[n]}\alpha_i\nabla_{\boldsymbol{w}}\mathcal{R}_i(\boldsymbol{w})$ and $\boldsymbol{H}_{\mathcal{R}}(\boldsymbol{w}) = \sum_{i\in[n]}\alpha_i\boldsymbol{H}_{\mathcal{R}_i}(\boldsymbol{w})$, we can get $\boldsymbol{w}^{t+1}$ from $\boldsymbol{w}_i^{t+1}$ via the following equation, which we call second-order aggregation,

$$\boldsymbol{w}^{t+1} = \boldsymbol{H}_{\mathcal{R}}(\boldsymbol{w}^t)^{-1}\sum_{i\in[n]}\alpha_i\boldsymbol{H}_{\mathcal{R}_i}(\boldsymbol{w}^t)\boldsymbol{w}_i^{t+1} \tag{8}$$

Specifically, to implement second-order aggregation, in each round, the local clients first optimize the model locally for several epochs. Then we compute the Hessian matrices for each local model and send them to the server for aggregation. Note that sending the Hessian takes a communication cost being quadratic to the size of the weight. In real practice, the predictive head is usually small, e.g., a linear layer with hundreds of neurons. Thus it is acceptable to aggregate the Hessian matrix of the head's parameters.

In the following, we provide two instances of our second-order aggregation with a linear head.

**1. Linear Regression** where $\mathcal{R}_i(\boldsymbol{w}) = \frac{1}{L_i}\sum_{j=1}^{L_i}(\boldsymbol{w}^\top\boldsymbol{x}_i^j - y^j)^2$ is quadratic itself. Thus the second order Taylor expansion of the objective itself, i.e., $\mathcal{J}_i(\boldsymbol{w}) = \mathcal{R}_i(\boldsymbol{w})$. In this case, $\boldsymbol{H}_{\mathcal{R}_i}(\boldsymbol{w}) = \boldsymbol{X}_i^\top\boldsymbol{X}_i$ where $\boldsymbol{X}_i = [\boldsymbol{x}_i^1, \cdots, \boldsymbol{x}_i^{L_i}]^\top$ is the data matrix of client $i$.

**2. Binary Classification** where $\mathcal{R}_i(\boldsymbol{w}) = -\frac{1}{L_i}\sum_{j=1}^{L_i}y_i^j\log\sigma(\boldsymbol{w}^\top\boldsymbol{x}_i^j) + (1-y_i^j)\log(1-\sigma(\boldsymbol{w}^\top\boldsymbol{x}_i^j))$. $\sigma$ is the sigmoid function. Let $\mu_i^j \triangleq \sigma(\boldsymbol{w}^\top\boldsymbol{x}_i^j)$ denote model's output. The gradient and the Hessian are, $\nabla_{\boldsymbol{w}}\mathcal{R}_i(\boldsymbol{w}) = \frac{1}{L_i}\sum_j(\mu_i^j - y_i^j)\boldsymbol{x}_i^j = \frac{1}{L_i}\boldsymbol{1}^\top\text{diag}(\boldsymbol{\mu}_i - \boldsymbol{y}_i)\boldsymbol{X}_i^\top$ and $\boldsymbol{H}_{\mathcal{R}_i}(\boldsymbol{w}) = \frac{1}{L_i}\boldsymbol{X}_i^\top\boldsymbol{S}\boldsymbol{X}_i$ where $\boldsymbol{S} \triangleq \text{diag}(\mu_i^1(1-\mu_i^1), \cdots, \mu_i^{L_i}(1-\mu_i^{L_i}))$. Similar formulas can be derived for the multiclass classification. Please refer to the text book (Murphy, 2022) for the exact equations.

**Remark.** In practice, when the dimension of $\boldsymbol{w}$ is larger than the number of samples of a certain domain, the Hessian may have small singular values which cause numerical instability. To mitigate this issue, one can either directly set the representation dimension $k$ to some smaller number or add a (fully-connected) projection layer on top of a pretrained encoder to compress the representations to a lower dimensional space.

### 4.4 Theoretical Result of FedDAR

For a simplified linear regression setting as discussed in domain-mixed FL (4) (cf. details in Appendix A), we give below the sample complexity required for an adapted version of our algorithm (Algorithm 2 in the appendix) to enjoy linear convergence. Due to the space limit, we only provide an informal statement to highlight the result. The formal statement and the proof are deferred in the appendix.

**Theorem 4.1** (Sample complexity of FedDAR convergence in linear case (informal)). *Consider the linear setting for domain-mixed FL in (4). At each iteration, suppose that the number of samples used by each of $n$ clients to update the encoder, is $\tilde{\Omega}(\frac{dk^2}{n})$, and that the aggregate number of samples used in the update for the domain-specific heads, is $\tilde{\Omega}(k)$. Then, for a suitably chosen step size, the distance between the encoder $\mathbf{B_t}$ Algorithm 2 outputs and the true encoder $\mathbf{B}^*$ converges at a linear rate.*

**Remark.** As our algorithm converges linearly to the true encoder, the per-iteration sample complexity of our algorithm gives a good estimate of the overall sample complexity. Since we expect the output of the encoder to be significantly lower-dimensional than the input (i.e. $k \ll d$), our result indicates that Algorithm 2's sample complexity is dominated by $\tilde{\Omega}(\frac{d}{n})$, implying that the complexity reduces significantly as the number of clients $n$ increases. Moreover, a key implication of our result is the capacity for our algorithm to accommodate data imbalance across domains. We note that our approach requires $\Omega(dk^2)$ samples per iteration for the update of the shared representation $\boldsymbol{B} \in \mathbb{R}^{d \times k}$, whilst needing only $\Omega(k)$ samples per iteration for the update of each domain head. In particular, domains with more data can contribute disproportionately to the $\tilde{\Omega}(dk^2)$ samples required to learn the common representation, whilst domains with fewer data need only provide $\tilde{\Omega}(k)$ samples to update its domain head during the course of the algorithm. Whenever $k \ll d$, which we believe is a reasonable assumption for many practical applications (e.g. medical imaging), the requirement of $\tilde{\Omega}(k)$ samples per domain is relatively mild. Conversely, forgoing the shared representation structure would require each domain to learn a separate $d$-dimensional classifier, requiring $\tilde{\Omega}(d)$ samples per domain, which can pose a challenge in problems with domain data imbalance.

## 5 EXPERIMENTS

We validate our method's effectiveness on both synthetic and real datasets. We first experiment on the exact synthetic dataset described in our theoretical analysis to verify our theory. We then conduct experiments on a real dataset, FairFace (Kärkkäinen & Joo, 2019), with controlled domain distributions to investigate the robustness of our algorithm under different levels of heterogeneity. Finally, we compare our method with various baselines on a real federated learning benchmark, EXAM (Dayan et al., 2021) with real-world domain distributions. We also conduct extensive ablation studies on it to discern the contribution of each component of our method. Full details of experimental settings can be found in Appendix B.

### 5.1 SYNTHETIC DATA

We first run experiments on the linear regression problem analyzed in Appendix A. We generate (domain, data, label) samples as the following, $z_i \sim \mathcal{M}(\boldsymbol{\pi_i})$, $\boldsymbol{x}_i \sim \mathcal{N}(0, \boldsymbol{I}_d)$, $y_i \sim \mathcal{N}(\boldsymbol{w}^*_{z_i}{}^\top \boldsymbol{B}^{*\top} \boldsymbol{x}_i, \sigma)$ where $\sigma = 10^{-3}$ controls label observation errors, $\mathcal{M}(\boldsymbol{\pi_i})$ is a multinomial domain distribution with parameter $\boldsymbol{\pi_i} = [\pi_{i,1}, ..., \pi_{i,M}] \in \Delta^M$. The hyper-parameters of domain distributions $\boldsymbol{\pi_i}$ are drawn from a Dirichlet distribution, i.e., $\boldsymbol{\pi_i} \sim Dir(\alpha \boldsymbol{p})$, where $\boldsymbol{p} \in \Delta^M$ is a prior domain distribution over $M$ domains, and $\alpha > 0$ is a concentration parameter controlling the heterogeneity of domain distributions among clients. The largest domain distribution heterogeneity is achieved as $\alpha \to 0$ where each client contains data only from a single randomly selected domain. On the other hand, when $\alpha \to \infty$, all clients have identical domain distributions that are equal to the prior $\boldsymbol{p}$. We generate ground-truth representation $\boldsymbol{B}^* \in \mathbb{R}^{d \times k}$ and domain specific heads $\boldsymbol{w}^*_m, \forall m \in [M]$ by sampling and normalizing Gaussian matrices.

Figure 5.2 shows result of our experiments where we set $n = 100$ clients, $M = 5$ domains, feature dimension $k = 2$. We vary the number of training samples per client from 5 to 20. The result shows that FedDAR-SA, achieves four orders of magnitude smaller errors than all the baselines: (1) Local-Only where each client train a model using its own data; (2) FedAvg which learns a single shared model; (3) FedRep which learns shared representation and client-specific heads. (4)Separate FedAvg which trains separate models for each domain using FedAvg. The results demonstrate that our method overcomes the heterogeneity of domain distributions across clients. FedDAR-WA fails to converge under this setting, confirming the effectiveness of the proposed second-order aggregation.

### 5.2 REAL DATA WITH CONTROLLED DISTRIBUTION

**Dataset and Model.** We use FairFace (Kärkkäinen & Joo, 2019), a public face image dataset containing 7 race groups which are considered as the domains. Each image is labeled with one of 9 age groups and gender. We use the age label as the target to build a multi-class age classifier. We created an FL setting by dividing training data into $n$ clients without duplication. Each client has a domain distribution $\boldsymbol{\pi_i} \sim Dir(\alpha \boldsymbol{p})$ sampled from a Dirichlet distribution. The total number of samples at each client $L_i = 500$ is set to be the same in all experiments. We control the heterogeneity of domain distributions by altering $\alpha$. The label distributions are uniform for all the clients.

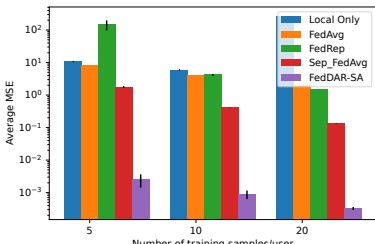

Figure 1: Performance under a different number of training samples per client, the error bars show the standard error from three independent runs.

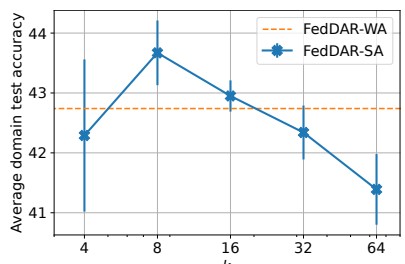

Figure 2: Age classification accuracy as a function of representation dimension $k$.

Table 1: Min, max and average test accuracy of age classification across 7 domains (*race groups*) on FairFace with number of clients $n = 5$, number of samples at each client $L_i = 500$

| Method | $\alpha = 0.1$ | | | $\alpha = 0.5$ | | | $\alpha = 1$ | | | $\alpha = 100$ | | |
|---|---|---|---|---|---|---|---|---|---|---|---|---|
| | Max | Min | Avg | Max | Min | Avg | Max | Min | Avg | Max | Min | Avg |
| Seperate FedAvg | 41.8±2.2 | 4.1±1.3 | 19.3±0.5 | 42.4±0.6 | 9.7±0.6 | 23.5±1.2 | 41.4±0.7 | 9.3±0.8 | 23.4±0.8 | 41.4±0.4 | 8.5±1.6 | 22.9±1.0 |
| FedAvg (w/o reweighting) | 43.4±1.7 | 37.1±1.1 | 40.0±0.6 | 45.7±0.5 | 38.8±0.2 | 41.6±0.3 | 45.2±0.8 | 38.4±0.8 | 41.1±0.4 | 44.0±0.7 | 38.6±0.7 | 40.8±0.7 |
| FedAvg(w/ reweighting) | 42.8±0.6 | 37.7±0.8 | 39.9±0.2 | 45.8±0.8 | 39.2±0.2 | 41.6±0.3 | 44.9±0.5 | 38.3±0.5 | 40.9±0.3 | 43.3±1.3 | 38.4±1.5 | 40.5±1.3 |
| FedAvg + Multi-head | 46.8±0.9 | 32.4±2.8 | 39.8±1.0 | 49.1±1.0 | 34.9±1.4 | 40.0±1.0 | **51.1**±0.4 | 34.7±0.2 | 40.3±0.6 | **49.6**±0.3 | 36.4±0.9 | 39.8±0.7 |
| FedDAR-WA | 46.5±1.9 | 34.0±1.9 | 40.5±0.3 | **49.9**±1.2 | 40.0±0.2 | 42.9±0.2 | 49.2±0.3 | 40.0±0.6 | 42.7±0.5 | 48.6±0.5 | 40.2±0.5 | 42.5±0.4 |
| FedDAR-SA | **48.2**±1.0 | **39.3**±0.8 | **42.4**±0.4 | 50.0±0.3 | **40.6**±0.4 | **43.4**±0.1 | 49.0±0.5 | **41.2**±0.3 | **43.7**±0.5 | 49.0±0.3 | **41.2**±0.4 | **43.6**±0.5 |

**Implementation and Evaluation.** We use Imagenet(Deng et al., 2009) pre-trained ResNet-34 (He et al., 2016) for all experiments on this dataset. All the methods are trained for $T = 100$ communication rounds. We use Adam optimizer with a learning rate of $1 \times 10^{-4}$ for the first 60 rounds and $1 \times 10^{-5}$ for the last 40 rounds.

**Metrics and Results.** Our evaluation metrics are the classification accuracy on the whole validation set of FairFace for each race group. We don't have extra local validation set for each client since we assume the data distribution within each domain is consistent across the clients. In Table 5.2, we report the accuracy averaged over the final 10 rounds of communication following the common practice (Collins et al., 2021). The result shows our FedDAR achieved the best performance compared with the baselines. Note that FedAvg + Multi-head also uses Equation 5 as objective for fair comparison.

**Effect of $k$.** The limitation of using FedDAR-SA instead of FedDAR-WA is the need of tuning the dimension of representation $k$. Figure 5.2 shows results of the average domain test accuracy with different $k$. We can see that FedDAR-SA can achieve better accuracy with a properly chosen $k$. We use $k = 8$ for all results with FedDAR-SA in Table 5.2.

**Robustness to Varying Levels of Heterogeneity.** From the result with various $\alpha$, we can observe that the performance of FedDAR-SA is very stable no matter how heterogeneous the domain mixtures are. However, the baselines' accuracy decrease when $\alpha$ becomes smaller.

## 5.3 Real Data with Real-World Data Distribution

**Dataset and Model.** We use the EXAM dataset (Dayan et al., 2021), a large-scale, real-world healthcare FL study. We use part of the dataset including 6 clients with a total of 7,681 cases. We use race groups as domains. The dataset is collected from suspected COVID-19 patients at the visit of the emergency department (ED), including both Chest X-rays (CXR) and electronic medical records (EMR). We adopt the same data preprocessing procedure and the model as (Dayan et al., 2021). Our task is to predict whether the patient received oxygen therapy higher than high-flow oxygen in 72 hours which indicates severe symptoms.

**Baselines.** (1) methods that learn one global model, FedAvg(McMahan et al., 2017a), FedProx(Li et al., 2020), FedMinMax(Papadaki et al., 2021) along with their local fine-tuned variants; (2) train $M$ separate models with FedAvg; (3) train one global model with FedAvg first, then fine-tune on $M$ domains separately with FedAvg; (4) client-wise personalized FL approaches, FedRep(Collins et al., 2021), FedPer(Arivazhagan et al., 2019), LG-Fedavg(Liang et al., 2020), FedBN(Li et al., 2021b).

**Implementation and Evaluation.** We apply 5-fold cross-validation. All the models are trained for $T = 20$ communication rounds with Adam optimizer and a learning rate of $10^{-4}$. The models are evaluated by aggregating predictions on the local validation sets and then calculating the area under curve (AUC) for each domain. We also report the AUCs averaged on clients' local validation set.

Table 3: AUCs result on EXAM dataset with the domain being *race group*. Numbers are the means and standard deviations of metrics from 5-fold cross-validation.

| Methods | White | Black | Asian | Latino | Other | Min | Avg | Client Avg |
|---|---|---|---|---|---|---|---|---|
| Local | .761±.023 | .815±.055 | .838±.039 | .889±.076 | .840±.038 | .759±.026 | .829±.032 | .795±.023 |
| separate FedAvg | .796±.022 | .694±.015 | .788±.047 | .649±.133 | .826±.046 | .606±.080 | .751±.026 | .759±.027 |
| FedAvg | .830±.027 | .854±.045 | .887±.022 | .834±.102 | .900±.038 | .773±.049 | .861±.019 | .856±.020 |
| FedAvg + FT | .783±.044 | .835±.025 | .892±.015 | .817±.136 | .892±.048 | .727±.093 | .844±.024 | .845±.016 |
| FedAvg + separate FT | .832±.032 | .846±.043 | .903±.025 | .869±.099 | .911±.026 | .784±.054 | .872±.017 | .863±.024 |
| FedProx | .834±.017 | .864±.056 | .903±.035 | .880±.085 | .912±.030 | .808±.030 | .879±.023 | .868±.012 |
| FedProx + FT | .806±.023 | .842±.049 | .910±.025 | .925±.085 | .898±.031 | .798±.025 | .876±.010 | .858±.014 |
| FedMinMax | .839±.027 | .867±.054 | .894±.039 | .916±.053 | .903±.034 | .823±.032 | .884±.020 | .872±.016 |
| FedBN | .787±.027 | .840±.063 | .883±.039 | .867±.090 | .852±.043 | .766±.013 | .846±.020 | .856±.010 |
| FedRep | .837±.020 | .869±.050 | .888±.042 | .913±.083 | .910±.028 | .812±.028 | .884±.025 | .867±.013 |
| FedPer | .835±.025 | .865±.073 | .909±.037 | .916±.036 | .911±.031 | .813±.047 | .887±.021 | .873±.011 |
| LG-FedAvg | .830±.029 | .858±.052 | .906±.032 | .902±.050 | .903±.033 | .814±.034 | .880±.019 | .867±.017 |
| FedDAR-WA | **.884±.007** | **.896±.017** | .902±.034 | **.952±.041** | .928±.022 | **.872±.015** | .912±.004 | .898±.006 |
| FedDAR-SA | **.888±.004** | **.895±.038** | **.928±.032** | .939±.046 | **.948±.016** | .868±.020 | **.919±.014** | **.912±.001** |

**Average Performance Across Domains and Clients.** Table 3 shows the average of AUCs across domains and clients. We can see that our methods, both FedDAR-WA and FedDAR-SA, achieve significantly better performance than all the baselines under both domain-wise and client-wise metrics. The gap between our domain-wise personalized approach and other client-wise personalized baselines shows the validity of learning domain-wise personalized models facing diversity across domains. The reason that fine-tuning methods induce worse results is mainly because of the imbalanced label distribution. Each local training dataset doesn't have enough positive cases to do proper fine-tuning.

**Fairness Across Domains.** The AUCs of each specific domain in Table 3, show that our proposed FedDAR method uniformly increases the AUC for each domain. The column of the minimum AUC among domains also verifies that our method indeed improves the fairness across the domains.

**Ablation Studies.** i) *re-weighting* (RW): First two rows in Table 2 shows adding sample re-weighting significantly improves the fairness across the domains. The minimum AUC among domains is improved by a large margin ($> 0.05$); ii) *multi-head* (MH), *domain as input feautre* (DI) and *alternating update* (Alter): Comparing three blocks in Table 2, we see that adding multi-head alone does not improve results. We conjecture that alternating update prevents the overfitting of the heads with limited samples. This is also shown by the result in Table 5.2, where FedAvg+MH tends to perform badly on certain underrepresented domains especially when domain distributions are highly heterogeneous ($\alpha$ is small). Meanwhile, using domain labels directly as feature input is not as good as multi-head, and not compatible with alternating update;

Table 2: Ablation results of different components' contribution in FedDAR.

| RW | MH | DI | Alter | Proj | AGG | Domain Avg / Min | Client Avg |
|---|---|---|---|---|---|---|---|
|  |  |  |  |  | N/A | .861 / .773 | .856 |
| ✓ |  |  |  |  | N/A | .881 / .824 | .873 |
| ✓ |  | ✓ |  |  | N/A | .880 / .825 | .866 |
| ✓ | ✓ |  |  |  | WA | .885 / .834 | .870 |
| ✓ | ✓ |  |  | ✓ | WA | .877 / .817 | .870 |
| ✓ | ✓ |  |  | ✓ | SA | .878 / .826 | .871 |
| ✓ |  | ✓ | ✓ |  | N/A | .867 / .806 | .852 |
| ✓ | ✓ |  | ✓ |  | WA | .912 / .872 | .898 |
| ✓ | ✓ |  | ✓ | ✓ | WA | **.918 / .863** | .904 |
| ✓ | ✓ |  | ✓ | ✓ | SA | **.919 / .868** | **.912** |

iii) *projection* (Proj) and *aggregation method* (AGG): Results in Table 2 shows that using second-order aggregation with the projection of the features gives the best result.

## 6 CONCLUSIONS

We propose a novel personalized federated learning framework that assumes the mixture of domain data distribution. Our approach, FedDAR, achieves a balanced performance across domains by learning a global representation and domain-specific heads, despite the heterogeneity of domain distributions across clients. Our method is effective, as supported by both theoretical and empirical justifications. It has been tested on face recognition and medical imaging FL datasets and can be easily extended to other complicated tasks. However, our method has some limitations: i) it requires the domain information for all samples; ii) it does not consider heterogeneity of label distributions; iii) it has a potentially expensive communication cost caused by sending Hessian matrices, especially when the output dimension is big. We plan to address these limitations in future work, along with other research directions such as improving fairness across domains and exploring the setting where domains are structured, hierarchical, continuously indexed (Wang et al., 2020; Nasery et al., 2021) or multi-dimensional (characterized by multiple factors).

ACKNOWLEDGMENTS

This work has been supported by NIH 1R01HL159183.

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

## A    FedDAR for Linear Representation

### A.1    Setup

We retain the setup for linear regression considered at the start of Section 3.1. We additionally define $\boldsymbol{W}^* \triangleq [\boldsymbol{w}_1^*, \cdots, \boldsymbol{w}_M^*]^\top \in \mathbb{R}^{M \times k}$ as the concatenation of domain specific heads. For notational convenience, we let $(\boldsymbol{x}_{i,m}, y_{i,m})$ denote an (input, output) sample coming from client $i$ and the $m$-th domain. To measure the distance between any two matrices $\boldsymbol{A}, \boldsymbol{B}$ with the same dimensions, we use the *principal angle distance* (Golub & Van Loan, 2013), given by $\text{dist}(\boldsymbol{A}, \boldsymbol{B}) \triangleq \|\boldsymbol{A}_\perp^\top \boldsymbol{B}\|_2$, where $\boldsymbol{A}_\perp$ denotes a matrix whose columns form a basis for the orthogonal complement of the range of $\boldsymbol{A}$. To simplify the analysis, we further make the following assumptions.

**Assumption A.1** (Sub-Gaussianilty). For each $m \in [M]$ and $i \in [n]$, the samples $\boldsymbol{x}_{i,m} \in \mathbb{R}^d$ are independent, mean zero, have covariance $\boldsymbol{I}_d$, and has subgaussian norm 1, i.e. for every $\boldsymbol{v} \in \mathbb{R}^d$, $\mathbb{E}[\exp(\boldsymbol{v}^\top \boldsymbol{x}_{i,m})] \leq \exp(\|\boldsymbol{v}\|^2/2)$.

**Assumption A.2** (Domain diversity). Let $\sigma_{\min,*} \triangleq \sigma_{\min}(\frac{1}{\sqrt{M}}\boldsymbol{W}^*)$, i.e., $\sigma_{\min,*}$ is the minimum singular value of the head matrix. Then $\sigma_{\min,*} > 0$.

**Assumption A.3** (Ground truth normalization). The true domain parameters satisfy $\frac{1}{2}\sqrt{k} \leq \|\boldsymbol{w}_m^*\| \leq \sqrt{k}$ for each $m \in [M]$, and $\boldsymbol{B}^*$ has orthonormal columns.

All the above assumptions aim to simplify the theoretical analysis whilst only imposing mild constraints on the data distribution and the parameters of the target functions. Similar assumptions have also been adapted in prior work (Collins et al., 2021).

### A.2    FedDAR Adapted to Linear Regression

We analyze an adapted version of our FedDAR algorithm. Since the linear regression problem has an analytic solution, to ease analysis, we update the heads $\{\boldsymbol{w}_m\}_{m=1}^M$ at the server in closed form using local gradient information. Meanwhile, we update the representation $\boldsymbol{B}$ by taking a step using the averaged local gradients. Algorithm 2 shows the procedure of this adapted version.

**The local objective**    for $i$-th client in $m$-th domain at $t$-th iteration, $f_{i,m}^t(\boldsymbol{w}_m, \boldsymbol{B}^t)$ is defined as the following,

$$f_{i,m}^t(\boldsymbol{w}_m, \boldsymbol{B}^t) \triangleq \frac{1}{2} \sum_{j=1}^{L_{i,m}^t} (y_{i,m}^j - \boldsymbol{w}_m^\top \boldsymbol{B}^\top \boldsymbol{x}_{i,m}^j)^2,$$

where $L_{i,m}^t$ is the number of samples from domain $m$ at client $i$. We assume in each iteration the data points $\{\boldsymbol{x}_{i,m}^j, y_{i,m}^j\}_{j \in [L_{i,m}^t]}$ are all newly sampled from the distribution. We denote $L = \sum_m L_{i,m}^t$. Note that since the objective function has a quadratic form, thus its gradient w.r.t either $\boldsymbol{w}_m$ or $\boldsymbol{B}$ has a linear form of $\boldsymbol{A}_{i,m}\boldsymbol{w}_m - \boldsymbol{a}_{i,n}$ or $\boldsymbol{C}_{i,m}\boldsymbol{B} - \boldsymbol{c}_{i,m}$ which we write down explicitly in Appendix B. After every global update of the representation $\boldsymbol{B}$, we apply an additional QR decomposition to normalize it to be column-wise orthogonal.

### A.3    Convergence Analysis

We first provide a brief proof sketch. Overall, our approach largely follows that in Collins et al. (2021), with a few differences needed to handle the spreading of a domain's data across different clients. We note that we also tightened the analysis compared with Collins et al. (2021), such that each domain only needs $O(k)$ samples as opposed to $O(k^2)$ samples as in Collins et al. (2021) (where the requirement is for each client to have $O(k^2)$ samples since they considered the case where each client has a separate head). This can yield a significant improvement when $k$ is moderately large and there is data imbalance.

1. First, in Lemma A.5, we show that our estimated weight matrix $W^{t+1} \in \mathbb{R}^{M \times k}$ (which is our estimation at time $t+1$ of the true domain weights matrix $\boldsymbol{W}^*$) satisfies the relationship
$$W^{t+1} = \boldsymbol{W}^*(\boldsymbol{B}^*)^\top \boldsymbol{B}^t + F_t,$$

---

**Algorithm 2** FEDDAR for linear regression

---

**Input:** Step size $\eta$; number of rounds $T$

**Client initialization:** each agent $i \in [n]$ collects $L^0$ samples, and sends $\boldsymbol{Z}_i := \sum_{i=1}^{L^0} (y_i^{0,j})^2 \boldsymbol{x}_i^{0,j} (\boldsymbol{x}_i^{0,j})^\top$ to the server.

**Server initialization:** finds $\boldsymbol{UDU}^\top \leftarrow$ rank-k SVD($\frac{1}{nL^0} \sum_i^n \boldsymbol{Z}_i$); sets $\boldsymbol{B}^0 \leftarrow \boldsymbol{U}$.

**for** $t = 0, 1, \ldots, T$ **do**

  Server sends current $\boldsymbol{B}^t$ to clients.

  **Client computation for $\boldsymbol{W}^{t+1}$:**

  **for** client $i \in [n]$ **do**

    Selects $L$ new samples $\{(\boldsymbol{x}_i^j, y_i^j)\}$.

    Computes $\nabla_{\boldsymbol{w}_m} f_{i,m}^t(\boldsymbol{w}_m, \boldsymbol{B}^t) = \boldsymbol{A}_{i,m}^t \boldsymbol{w}_m - \boldsymbol{a}_{i,m}^t$ for each domain $m \in [M]$.

    Sends $(\boldsymbol{A}_{i,m}^t, \boldsymbol{a}_{i,m}^t, L_{i,m}^t)$ back to server.

  **end for**

  **Server update for $\boldsymbol{W}^{t+1}$:**

  Server chooses $\boldsymbol{w}_m^{t+1} \in \left\{ \boldsymbol{w}_m \in \mathbb{R}^k : \nabla_{\boldsymbol{w}_m} \left( \frac{1}{\sum_i L_{i,m}^t} \sum_{i=1}^n f_{i,m}^t(\boldsymbol{w}_m, \boldsymbol{B}^t) \right) = 0 \right\}, \forall m \in$ $[M]$, i.e., $\boldsymbol{w}_m^{t+1}$ that satisfies $(\sum_i \boldsymbol{A}_{i,m}^t)\boldsymbol{w}_m^{t+1} = \sum_i \boldsymbol{a}_{i,m}^t$.

  Sends $\boldsymbol{W}^{t+1} = [\boldsymbol{w}_1, \cdots, \boldsymbol{w}_M]^\top \in \mathbb{R}^{M \times k}$ to clients.

  **Client computation for $\boldsymbol{B}^{t+1}$:**

  **for** client $i \in [n]$ **do**

    Selects $L$ new samples $\{\boldsymbol{x}_i^j, y_i^j\}$.

    Computes $\nabla_{\boldsymbol{B}} f_{i,m}^{t'}(\boldsymbol{w}_m^{t+1}, \boldsymbol{B}^t) = \boldsymbol{C}_{i,m}^t \boldsymbol{B}^t - \boldsymbol{c}_{i,m}^t$ for each $m \in [M]$.

    Sends $(\nabla_{\boldsymbol{B}} f_{i,m}^{t'}(\boldsymbol{w}_m^{t+1}, \boldsymbol{B}^t), L_{i,m}^{t'})$ back to server.

  **end for**

  **Server update for $\boldsymbol{B}^{t+1}$:**

  Server computes $\tilde{\boldsymbol{B}}^{t+1} \leftarrow \boldsymbol{B}^t - \eta \frac{1}{m} \sum_{m=1}^M \frac{1}{\sum_i L_{i,m}^{t'}} \sum_{i=1}^n \nabla_{\boldsymbol{B}} f_{i,m}^{t'}(\boldsymbol{w}_m^{t+1}, B)$.

  Server performs QR decomposition $\hat{\boldsymbol{B}}^{t+1}, \boldsymbol{R}^{t+1} = \text{QR}(\tilde{\boldsymbol{B}}^{t+1})$.

  Server updates $\boldsymbol{B}^{t+1} \leftarrow \hat{\boldsymbol{B}}^{t+1}$.

**end for**

---

where $\boldsymbol{B}^t$ is our estimation of the true (high to low dimensional) representation embedding $\boldsymbol{B}^*$ at time $t$, and $F_t$ is an error term that we can show to be bounded (sufficiently small) in terms of $\text{dist}(\boldsymbol{B}^t, \boldsymbol{B}^*)$, with the scale of the error depending on the (random) number of samples $L_m$ seen at time $t$ for each domain $m \in [M]$. Bounding $\|F_t\|$ in our setting requires some care since the samples for each domain are spread over many clients. We refer the reader to Section A.4.2 for the details.

2. Second, we show in Section A.4.3 that the update for $\boldsymbol{B}^t$ satisfies the relationship (see equation 21)

$$\boldsymbol{B}^{t+1} = \boldsymbol{B}^t - \frac{\eta}{M} \left( (Q^t)^\top \right) W^{t+1} - \frac{\eta}{M} H_Q,$$

where $Q^t := W^{t+1}(\boldsymbol{B}^t)^\top - W^*(\boldsymbol{B}^*)^\top$, and $H_Q$ denotes an error term which can be shown to be bounded (sufficiently small) in terms of $\text{dist}(\boldsymbol{B}^t, \boldsymbol{B}^*)$. Further simplifying, we have

$$\text{dist}(\boldsymbol{B}^{t+1}, \boldsymbol{B}^*) = (\boldsymbol{B}_\perp^*)^\top \boldsymbol{B}^{t+1} = (\boldsymbol{B}_\perp^*)^\top \left( \boldsymbol{B}^t - \frac{\eta}{M} \left( \boldsymbol{B}^t(W^{t+1})^\top - \boldsymbol{B}^*(\boldsymbol{W}^*)^\top \right) W^{t+1} - \frac{\eta}{M} H_Q \right),$$

$$= \text{dist}(\boldsymbol{B}^t, \boldsymbol{B}^*) - \frac{\eta}{M}(\boldsymbol{B}_\perp^*)^\top \boldsymbol{B}^t (W^{t+1})^\top W^{t+1} - \frac{\eta}{M}(\boldsymbol{B}_\perp^*)^\top H_Q$$

$$\leq \text{dist}(\boldsymbol{B}^t, \boldsymbol{B}^*) - \frac{\eta}{M}\text{dist}(\boldsymbol{B}^t, \boldsymbol{B}^*)\sigma_{\min}^2(W^{t+1}) - \frac{\eta}{M}(\boldsymbol{B}_\perp^*)^\top H_Q$$

By upper bounding $\|H_Q\|$ in terms of $\text{dist}(\boldsymbol{B}^t, \boldsymbol{B}^*)$ and providing an appropriate lower bound on $\sigma_{\min}(W_{t+1}^\top W_{t+1})$, we can then show by picking $\eta$ sufficiently small and under other suitable assumptions, the quantity $\text{dist}(\boldsymbol{B}^t, \boldsymbol{B}^*)$ decays at a linear rate (see Equation 29). Again, a difference from the analysis in Collins et al. (2021) is our handling of bounding $\|H_Q\|$, since the samples for each sample are spread over different clients.

3. An issue created by the spreading of samples for a domain across different clients is how to pick an appropriate sample size such that the $\|F_t\|$ and $\|H_Q\|$ terms can be suitably bounded. In Lemma A.10, we prescribe a suitable sample size for each client such that each domain gets sufficient samples (with high probability) for the purposes of our analysis.

We now proceed with a detailed analysis.

We first present a theorem that states our adapted FedDAR(Algorithm 2) enjoys linear convergence. The theorem is followed by multiple remarks which highlight key detailed points of our convergence result.

**Theorem A.4** (Algorithm 2 convergence). *Define* $E_0 := 1 - \text{dist}^2(\boldsymbol{B}^0, \boldsymbol{B}^*)$, $\bar{\sigma}_{\max,*} := \sigma_{\max}\left(\frac{1}{\sqrt{M}}\boldsymbol{W}^*\right)$, $\bar{\sigma}_{\min,*} := \sigma_{\min}\left(\frac{1}{\sqrt{M}}\boldsymbol{W}^*\right)$. *Let* $\kappa := \frac{\bar{\sigma}_{\max,*}}{\bar{\sigma}_{\min,*}}$. *Suppose*

$$L \geq \tilde{\Omega}\left(\max\left\{\frac{dk^2\kappa^4}{nE_0^2}, \frac{k^2\kappa^4}{E_0^2 \min_{m\in[M]}(\sum_{i=1}^m \pi_{i,m})}\right\}\right). \tag{9}$$

*Then, for any $T$ and any* $\eta \leq 1/(4\bar{\sigma}_{\max,*}^2)$*, with probability at least* $1 - Te^{-80}$*,*

$$\text{dist}(\boldsymbol{B}^T, \boldsymbol{B}^*) \leq (1 - \eta E_0 \bar{\sigma}_{\min,*}^2/2)^{T/2}\text{dist}(\boldsymbol{B}^0, \boldsymbol{B}^*). \tag{10}$$

**Linear convergence speed:** The convergence of $\boldsymbol{B}^T$ to $\boldsymbol{B}^*$ is linear, assuming that (1) $\sigma_{\min}(\frac{1}{\sqrt{M}}\boldsymbol{W}^*) > 0$ and that (2) $1 - \eta E_0 \bar{\sigma}_{\min}^2 \in (0, 1)$.

**Initialization of $\boldsymbol{B}^0$:** For our convergence result to be meaningful, we need $\text{dist}(\boldsymbol{B}^0, \boldsymbol{B}^*)$ to be close to 0. We show in Appendix A that our algorithm's choice of initial $\boldsymbol{B}^0$ ensures that $\text{dist}(\boldsymbol{B}^0, \boldsymbol{B}^*)$ is close enough to 0 whilst preserving privacy. When the number of samples is uniform across the domains, this comes only at the cost of a logarithmic increase in sample complexity.

**Sample complexity:** The per-iteration sample complexity per client is $L$. We note that in the requirement for $L$ (9), we need that $L \geq \Omega(dk^2\kappa^4/n)$; this comes from the updates for $\boldsymbol{B}^t \in \mathbb{R}^{d\times k}$. While we expect that $d$ could be large, a large number of clients $n$ helps to mitigate the increase in sample complexity arising from $d$. We also need $L \geq \Omega(k^2\kappa^4 \sum_{i=1}^m \pi_{i,m})$ for every domain $m \in [M]$; this requirement comes from the updates for $\boldsymbol{w}_m^t$ for each of the $M$ domains.

## A.4 Proof of Theorem A.4

### A.4.1 Analysis for update of $W^{t+1}$

Since we are analyzing the update step for any iteration $t$, unless necessary we drop all $t$ superscripts. Let $L_m = \sum_{i=1}^n L_{i,m}$ denote the number of samples from domain $m \in [M]$ across the $n$ clients. Then, we can express $\nabla_{\boldsymbol{w}_m} \sum_{i=1}^n f_{i,m}(\boldsymbol{w}_m, \boldsymbol{B})$ as

$$\nabla_{\boldsymbol{w}_m} \sum_{i=1}^n f_{i,m}(\boldsymbol{w}_m, \boldsymbol{B}) = \sum_{i=1}^n \sum_{j=1}^{L_{i,m}} (\boldsymbol{w}_m^\top \boldsymbol{B}^\top \boldsymbol{x}_{i,m}^j - y_{i,m}^j)\boldsymbol{B}^\top \boldsymbol{x}_{i,m}^j.$$

Since

$$y_{i,m}^j = (\boldsymbol{w}_m^*)^\top (\boldsymbol{B}^*)^\top \boldsymbol{x}_{i,m}^j,$$

it follows that following Algorithm 2,

$$\underbrace{\left(\frac{1}{L_m}\sum_{i=1}^n \sum_{j=1}^{L_{i,m}} \left(\boldsymbol{B}^\top \boldsymbol{x}_{i,m}^j (\boldsymbol{x}_{i,m}^j)^\top \boldsymbol{B}\right)\right)}_{G_m} \boldsymbol{w}_m^{t+1} = \frac{1}{L_m}\sum_{i=1}^n \sum_{j=1}^{L_{i,m}} \left(\boldsymbol{B}^\top \boldsymbol{x}_{i,m}^j (\boldsymbol{x}_{i,m}^j)^\top \boldsymbol{B}^*\right)\boldsymbol{w}_m^*. \tag{11}$$

Reexpressing, assuming $G_m$ is invertible, we have

$$\boldsymbol{w}_m^{t+1} = \boldsymbol{B}^\top \boldsymbol{B}^* \boldsymbol{w}_m^* + \left(G_m^{-1}\left(\frac{1}{L_m}\sum_{i=1}^n \sum_{j=1}^{L_{i,m}} \left(\boldsymbol{B}^\top \boldsymbol{x}_{i,m}^j (\boldsymbol{x}_{i,m}^j)^\top \boldsymbol{B}^*\right)\boldsymbol{w}_m^*\right) - \boldsymbol{B}^\top \boldsymbol{B}^* \boldsymbol{w}_m^*\right) \tag{12}$$

Intuitively, assuming $L_m$ is large enough,

$$\frac{1}{L_m} \sum_{i=1}^{n} \sum_{j=1}^{L_{i,m}} \boldsymbol{x}_{i,m}^j (\boldsymbol{x}_{i,m}^j)^\top \approx I_d.$$

Hence,

$$G_m^{-1} \left( \frac{1}{L_m} \sum_{i=1}^{n} \sum_{j=1}^{L_{i,m}} \left( \boldsymbol{B}^\top \boldsymbol{x}_{i,m}^j (\boldsymbol{x}_{i,m}^j)^\top \boldsymbol{B}^* \right) \boldsymbol{w}_m^* \right) \approx \boldsymbol{B}^\top \boldsymbol{B}^* \boldsymbol{w}_m^*.$$

This then implies that

$$W^{t+1} = W^* (\boldsymbol{B}^*)^\top \boldsymbol{B} + F, \tag{13}$$

where the $m$-th row of $F$ is

$$F_m^\top := \left( G_m^{-1} \left( \frac{1}{L_m} \sum_{i=1}^{n} \sum_{j=1}^{L_{i,m}} \left( \boldsymbol{B}^\top \boldsymbol{x}_{i,m}^j (\boldsymbol{x}_{i,m}^j)^\top \boldsymbol{B}^* \right) \boldsymbol{w}_m^* \right) - \boldsymbol{B}^\top \boldsymbol{B}^* \boldsymbol{w}_m^* \right)^\top.$$

Note the similarity of equation 18 to (17) in (Collins et al., 2021). Following a similar analysis as (Collins et al., 2021), we should also be able to bound the Frobenius norm of $F$ in terms of $\mathrm{dist}(\boldsymbol{B}, \boldsymbol{B}^*)$.

Below, we formalize the argument. First, we have the following lemma.

**Lemma A.5** (Update for $W^{t+1}$). *For each time $t$, let $L_m^t := \sum_{i=1}^{n} L_{i,m}^t$ denote the number of samples from domain $m \in [M]$ across the $n$ clients at time $t$. For convenience, we drop the time index unless absolutely necessary. We define the terms*

$$X_m := \frac{1}{L_m} \sum_{i=1}^{n} \sum_{j=1}^{L_{i,m}} \boldsymbol{x}_{i,m}^j (\boldsymbol{x}_{i,m}^j)^\top, \quad G_m := \frac{1}{L_m} \sum_{i=1}^{n} \sum_{j=1}^{L_{i,m}} \left( \boldsymbol{B}^\top \boldsymbol{x}_{i,m}^j (\boldsymbol{x}_{i,m}^j)^\top \boldsymbol{B} \right).$$

*Then, assuming that $G_m$ is invertible, the update for $W$ takes the form*

$$W^{t+1} = W^* (\boldsymbol{B}^*)^\top \boldsymbol{B} + F, \tag{14}$$

*where the $m$-th row of $F$ is*

$$F_m^\top := \left( G_m^{-1} \left( \boldsymbol{B}^\top X_m (I - \boldsymbol{B}\boldsymbol{B}^\top) \boldsymbol{B}^* \right) \boldsymbol{w}_m^* \right)^\top. \tag{15}$$

*Proof.* We can express $\nabla_{\boldsymbol{w}_m} \sum_{i=1}^{n} f_{i,m}(\boldsymbol{w}_m, \boldsymbol{B})$ as

$$\nabla_{\boldsymbol{w}_m} \sum_{i=1}^{n} f_{i,m}(\boldsymbol{w}_m, \boldsymbol{B}) = \sum_{i=1}^{n} \sum_{j=1}^{L_{i,m}} (\boldsymbol{w}_m^\top \boldsymbol{B}^\top \boldsymbol{x}_{i,m}^j - y_{i,m}^j) \boldsymbol{B}^\top \boldsymbol{x}_{i,m}^j.$$

Since

$$y_{i,m}^j = (\boldsymbol{w}_m^*)^\top (\boldsymbol{B}^*)^\top \boldsymbol{x}_{i,m}^j,$$

it follows that following Algorithm 2,

$$\underbrace{\left( \frac{1}{L_m} \sum_{i=1}^{n} \sum_{j=1}^{L_{i,m}} \left( \boldsymbol{B}^\top \boldsymbol{x}_{i,m}^j (\boldsymbol{x}_{i,m}^j)^\top \boldsymbol{B} \right) \right)}_{G_m} \boldsymbol{w}_m^{t+1} = \frac{1}{L_m} \sum_{i=1}^{n} \sum_{j=1}^{L_{i,m}} \left( \boldsymbol{B}^\top \boldsymbol{x}_{i,m}^j (\boldsymbol{x}_{i,m}^j)^\top \boldsymbol{B}^* \right) \boldsymbol{w}_m^*. \tag{16}$$

Reexpressing, assuming $G_m$ is invertible, we have

$$\boldsymbol{w}_m^{t+1} = \boldsymbol{B}^\top \boldsymbol{B}^* \boldsymbol{w}_m^* + \left( G_m^{-1} \left( \frac{1}{L_m} \sum_{i=1}^{n} \sum_{j=1}^{L_{i,m}} \left( \boldsymbol{B}^\top \boldsymbol{x}_{i,m}^j (\boldsymbol{x}_{i,m}^j)^\top \boldsymbol{B}^* \right) \boldsymbol{w}_m^* \right) - \boldsymbol{B}^\top \boldsymbol{B}^* \boldsymbol{w}_m^* \right). \tag{17}$$

This then implies that

$$W^{t+1} = W^*(\boldsymbol{B}^*)^\top \boldsymbol{B} + F, \tag{18}$$

where the $m$-th row of $F$ is

$$
\begin{aligned}
F_m^\top &:= \left( G_m^{-1} \left( \frac{1}{L_m} \sum_{i=1}^n \sum_{j=1}^{L_{i,m}} \left( \boldsymbol{B}^\top \boldsymbol{x}_{i,m}^j (\boldsymbol{x}_{i,m}^j)^\top \boldsymbol{B}^* \right) \boldsymbol{w}_m^* \right) - \boldsymbol{B}^\top \boldsymbol{B}^* \boldsymbol{w}_m^* \right)^\top \\
&= \left( G_m^{-1} \boldsymbol{B}^\top X_m \boldsymbol{B}^* \boldsymbol{w}_m^* - G_m^{-1} G_m \boldsymbol{B}^\top \boldsymbol{B}^* \boldsymbol{w}_m^* \right)^\top \\
&= \left( G_m^{-1} \boldsymbol{B}^\top X_m \boldsymbol{B}^* \boldsymbol{w}_m^* - G_m^{-1} \boldsymbol{B}^\top X_m \boldsymbol{B} \boldsymbol{B}^\top \boldsymbol{B}^* \boldsymbol{w}_m^* \right)^\top \\
&= \left( G_m^{-1} \left( \boldsymbol{B}^\top X_m (I - \boldsymbol{B}\boldsymbol{B}^\top) \boldsymbol{B}^* \right) \boldsymbol{w}_m^* \right)^\top.
\end{aligned}
$$

$\square$

### A.4.2 Bounding $\|F\|_F$

We will proceed to bound the Frobenius norm of $F$. We begin by showing that $G_m^{-1}$ exists and (both lower and upper) bounding its spectral norm.

**Lemma A.6.** *Let $L_{\min} := \min_{m \in [M]} L_m$. Let $\delta_k := \frac{10C\sqrt{k}\sqrt{\log(M)}}{\sqrt{L_{\min}}}$ for some absolute constant $C$. Suppose that $0 \le \delta_k < 1$. Then, with probability at least $1 - e^{-80k \log(M)}$, $G_m^{-1}$ exists for each $m \in [M]$, and*

$$\|G_m^{-1}\|_2 \le \frac{1}{1 - \delta_k} \quad \forall m \in [M].$$

*Proof.* Note that

$$G_m := \frac{1}{L_m} \sum_{i=1}^n \sum_{j=1}^{L_{i,m}} \left( \boldsymbol{B}^\top \boldsymbol{x}_{i,m}^j (\boldsymbol{x}_{i,m}^j)^\top \boldsymbol{B} \right).$$

Let $v_{i,m}^j := \boldsymbol{B}^\top \boldsymbol{x}_{i,m}^j$. Since $\boldsymbol{B}^\top \boldsymbol{B} = I$, it follows that each $v_{i,m}^j$ is i.i.d 1-subgaussian. Then, applying the same argument in Theorem 4.6.1 of Vershynin 2018, we have (cf. equation (4.22) in Vershynin 2018)

$$\sigma_{\min}(G_m) \ge 1 - \underbrace{C \left( \frac{\sqrt{k}}{\sqrt{L_m}} + \frac{z}{\sqrt{L_m}} \right)}_{\delta_{k,m}} \tag{19}$$

with probability at least $1 - e^{-z^2}$ for $z \ge 0$ and some absolute constant $C$, assuming that $0 \le \delta_{k,m} \le 1$. Consider the choice $z = 9\sqrt{k}\sqrt{\log(M)}$. Then,

$$\delta_{k,m} = C \left( \frac{\sqrt{k}}{\sqrt{L_m}} + \frac{9\sqrt{k}\log(M)}{\sqrt{L_m}} \right) \le 10C \frac{\sqrt{k}\log M}{\sqrt{L_m}} \le 10C \frac{\sqrt{k}\sqrt{\log M}}{\sqrt{L_{\min}}}.$$

Suppose we choose $L_{\min} \ge 1$ such that $\delta_{k,m} < 1$. Then, taking a union bound, with probability at least $1 - Me^{-z^2} = 1 - M \exp(-81k\} \log(M)) = 1 - \exp(-80k \log(M))$,

$$\sigma_{\min}(G_m) \ge 1 - \delta_{k,m} \ge 1 - \frac{10C\sqrt{k}\sqrt{\log M}}{\sqrt{L_{\min}}} > 0 \quad \forall m \in [M]. \tag{20}$$

Therefore, with probability at least $1 - \exp(-80k \log(M))$, $G_m^{-1}$ exists for every $m \in [M]$, and in addition,

$$\|G_m^{-1}\|_2 \le \frac{1}{1 - \delta_k} \quad \forall m \in [M].$$

$\square$

We next bound the operator norm of term $\boldsymbol{B}^\top X_m(I - \boldsymbol{B}\boldsymbol{B}^\top)\boldsymbol{B}^*$.

**Lemma A.7.** *Let* $L_{\min} := \min_{m \in [M]} L_m$. *Let* $\delta_k := \frac{10C\sqrt{k}\sqrt{\log M}}{\sqrt{L_{\min}}}$ *for some absolute constant* $C$. *Suppose* $L_{\min}$ *is such that* $0 \le \delta_k < 1$. *Then, with probability at least* $1 - e^{-50k \log M}$,

$$\|\boldsymbol{B}^\top X_m(I - \boldsymbol{B}\boldsymbol{B}^\top)\boldsymbol{B}^*\|_2 \le \text{dist}(\boldsymbol{B}^*, B)\delta_k.$$

*Proof.* We will use an $\epsilon$-net argument, similar to the proof of Theorem 4.6.1 in (Vershynin, 2018).

First, by Corollary 4.2.13 in (Vershynin, 2018), there exists an $1/4$-net $\mathcal{N}$ of the unit sphere $S^{k-1}$ with cardinality $\mathcal{N} \le 9^k$. Using Lemma 4.4.1 in (Vershynin, 2018), we have that

$$\|\boldsymbol{B}^\top X_m(I - \boldsymbol{B}\boldsymbol{B}^\top)\boldsymbol{B}^*\|_2 \le 2\max_{z \in \mathcal{N}}\left|\left\langle \left(\boldsymbol{B}^\top X_m(I - \boldsymbol{B}\boldsymbol{B}^\top)\boldsymbol{B}^*\right)z, z\right\rangle\right|.$$

To prove our result, by applying a union bound over $m \in [M]$, it suffices to show that with the probability at least $1 - e^{-100k^2 \log M}$,

$$\max_{z \in \mathcal{N}}\left|\left\langle\left(\boldsymbol{B}^\top X_m(I - \boldsymbol{B}\boldsymbol{B}^\top)\boldsymbol{B}^*\right)z, z\right\rangle\right| \le \frac{\delta_{k_m}}{2} \quad \forall m \in [M],$$

where we recall that

$$\delta_{k,m} = C\left(\frac{\sqrt{k}}{\sqrt{L_m}} + \frac{9\sqrt{k}\log(M)}{\sqrt{L_m}}\right) \le \delta_k.$$

We will assume that $\min_m L_m := L_{\min} \ge 1$ is chosen large enough such that $\delta_{k,m} \le 1$.

For a fixed $z \in S^{k-1}$, observe that

$$\left\langle\left(\boldsymbol{B}^\top X_m(I - \boldsymbol{B}\boldsymbol{B}^\top)\boldsymbol{B}^*\right)z, z\right\rangle = \frac{1}{L_m}\sum_{i=1}^{n}\sum_{j=1}^{L_{i,m}}\left\langle\left(B^\top \boldsymbol{x}_{i,m}^j(\boldsymbol{x}_{i,m}^j)^\top(I - \boldsymbol{B}\boldsymbol{B}^\top)\boldsymbol{B}^*\right)z, z\right\rangle$$

$$:= \frac{1}{L_m}\sum_{i=1}^{n}\sum_{j=1}^{L_{i,m}}(z^\top u_{i,m}^j)((v_{i,m}^j)^\top z),$$

where we defined $u_{i,m}^j := \boldsymbol{B}^\top \boldsymbol{x}_{i,m}^j$, and $v_{i,m}^j = (\boldsymbol{B}^*)^\top(I - \boldsymbol{B}\boldsymbol{B}^\top)\boldsymbol{x}_{i,m}^j$.

Since each $x_{i,m}^j$ is 1-subgaussian, $\|\boldsymbol{B}\|_2 = 1$, and $\|(I - \boldsymbol{B}\boldsymbol{B}^\top)\boldsymbol{B}^*\|_2 = \text{dist}(\boldsymbol{B}^*, \boldsymbol{B})$, it follows that $z^\top u_{i,m}^j$ is subgaussian with norm at most 1, and $(v_{i,m}^j)^\top z$ is subgaussian with norm at most $\text{dist}(\boldsymbol{B}^*, \boldsymbol{B})$. Thus, the random variable $\alpha_{i,m}^j := (z^\top u_{i,m}^j)((v_{i,m}^j)^\top z)$ (for a fixed unit $z$) is sub-exponential with sub-exponential norm at most $\text{dist}(\boldsymbol{B}^*, \boldsymbol{B})$. Moreover, note that $\alpha_{i,m}^j$ is mean-zero, since

$$\mathbb{E}[u_{i,m}^j(v_{i,m}^j)^\top] = \mathbb{E}[B^\top \boldsymbol{x}_{i,m}^j(\boldsymbol{x}_{i,m}^j)^\top(I - \boldsymbol{B}\boldsymbol{B}^\top)\boldsymbol{B}^*]$$
$$= \boldsymbol{B}^\top(I - \boldsymbol{B}\boldsymbol{B}^\top)\boldsymbol{B}^* = 0,$$

as $x_{i,m}^j$ is assumed to have identity covariance. Thus, the $\alpha_{i,m}^j$'s are i.i.d mean-zero subexponential variables each with subexponential norm at most $\text{dist}(\boldsymbol{B}^*, \boldsymbol{B})$. Hence, by Bernstein's inequality (cf. Corollary 2.8.3 in (Vershynin, 2018)),

$$\mathbb{P}\left(\left|\left\langle\left(\boldsymbol{B}^\top X_m(I - \boldsymbol{B}\boldsymbol{B}^\top)\boldsymbol{B}^*\right)z, z\right\rangle\right| \ge \frac{\delta_{k,m}\text{dist}(\boldsymbol{B}^*, \boldsymbol{B})}{2}\right)$$

$$= \mathbb{P}\left(\left|\frac{1}{L_m}\sum_{i=1}^{n}\sum_{j=1}^{L_{i,m}}\alpha_{i,m}^j\right| \ge \frac{\delta_{k,m}\text{dist}(\boldsymbol{B}^*, \boldsymbol{B})}{2}\right)$$

$$\le 2\exp\left(-c\min\left(\frac{\delta_{k,m}\text{dist}(\boldsymbol{B}^*, \boldsymbol{B})}{\text{dist}(\boldsymbol{B}^*, \boldsymbol{B})}, \left(\frac{\delta_{k,m}\text{dist}(\boldsymbol{B}^*, \boldsymbol{B})}{\text{dist}(\boldsymbol{B}^*, \boldsymbol{B})}\right)^2\right)L_m\right)$$

$$= 2\exp(-c\delta_{k,m}^2 L_m)$$

$$\le 2\exp(-cC^2(k + 81k\log(M)\log(M))).$$

Above we used the assumption that $\delta_{k,m} \leq 1$ to simplify the minimum operator in the exponent.

Taking a union bound over each $z \in \mathcal{N}$, it follows that

$$\mathbb{P}\left(\|\boldsymbol{B}^\top X_m(I - \boldsymbol{B}\boldsymbol{B}^\top)\boldsymbol{B}^*\|_2 \geq \delta_{k,m}\text{dist}(\boldsymbol{B}^*, \boldsymbol{B})\right)$$

$$\leq \mathbb{P}\left(2\max_{z \in \mathcal{N}}\left|\left\langle\left(\boldsymbol{B}^\top X_m(I - \boldsymbol{B}\boldsymbol{B}^\top)\boldsymbol{B}^*\right)z, z\right\rangle\right| \geq \delta_{k,m}\text{dist}(\boldsymbol{B}^*, \boldsymbol{B})\right)$$

$$\leq 2 \cdot 9^k \exp(-cC^2(k + 81k\log(M)))$$

$$\leq \exp(-51k\log M),$$

where the last inequality follows by picking $C$ large enough (but still it is an absolute constant). (Here the choice of 51 in the exponent is somewhat arbitrary; any choice smaller than 81 should work). By applying a union bound over the domains $m \in [M]$, this then completes our proof. □

We are now finally ready to bound $\|F\|_F$.

**Lemma A.8.** *Let* $L_{\min} := \min_{m \in [M]} L_m$. *Let* $\delta_k := \frac{10C\sqrt{k}\sqrt{\log(M)}}{\sqrt{L_{\min}}}$ *for some absolute constant* $C$. *Suppose that* $0 \leq \delta_k < 1$. *Then, with probability at least* $1 - 2e^{-50k\log(M)}$,

$$\|F\|_F \leq \frac{\delta_k}{1 - \delta_k}\text{dist}(\boldsymbol{B}^*, \boldsymbol{B})\|W^*\|_F.$$

*Proof.* By Lemma A.6 and Lemma A.7, we have that with probability at least $1 - 2e^{-50k\log M}$,

$$\left\|G_m^{-1}(\boldsymbol{B}^\top X_m(I - \boldsymbol{B}\boldsymbol{B}^\top)\boldsymbol{B}^*)\right\|_2 \leq \left\|G_m^{-1}\right\|_2\left\|\boldsymbol{B}^\top X_m(I - \boldsymbol{B}\boldsymbol{B}^\top)\boldsymbol{B}^*\right\|_2$$

$$\leq \frac{1}{1 - \delta_k}\left(\delta_k\text{dist}(\boldsymbol{B}^*, \boldsymbol{B})\right).$$

The proof then follows by recalling that the $m$-th row, $F_m^\top$, takes the form

$$F_m^\top = \left(G_m^{-1}\left(\boldsymbol{B}^\top X_m(I - \boldsymbol{B}\boldsymbol{B}^\top)\boldsymbol{B}^*\right)\boldsymbol{w}_m^*\right)^\top.$$

□

### A.4.3 ANALYSIS OF UPDATE FOR $B^{t+1}$

Similarly to (Collins et al., 2021), we define

$$Q^t = W^{t+1}(\boldsymbol{B}^t)^\top - (W^*)(\boldsymbol{B}^*)^\top.$$

Below, we drop the time index and use $\boldsymbol{B}, Q, W$ to denote $\boldsymbol{B}^t, Q^t$, and $W^{t+1}$ respectively. Based on algorithm 2, we have that

$$\tilde{\boldsymbol{B}}^{t+1} = \boldsymbol{B} - \frac{\eta}{M}\sum_{m=1}^M \frac{1}{L_m}\sum_{i=1}^n \sum_{j=1}^{L_{i,m}}(\boldsymbol{w}_m^\top \boldsymbol{B}^\top \boldsymbol{x}_{i,m}^j - y_{i,m}^j)\boldsymbol{x}_{i,m}^j \boldsymbol{w}_m^\top$$

$$= \boldsymbol{B} - \frac{\eta}{M}\sum_{m=1}^M \frac{1}{L_m}\sum_{i=1}^n \sum_{j=1}^{L_{i,m}}\left(\left\langle A_{i,m}^j, W\boldsymbol{B}^\top\right\rangle - \left\langle A_{i,m}^j, W^*(\boldsymbol{B}^*)^\top\right\rangle\right)(A_{i,m}^j)^\top W, \qquad A_{i,m}^j := e_m(\boldsymbol{x}_{i,m}^j)^\top$$

$$= \boldsymbol{B} - \frac{\eta}{M}\sum_{m=1}^M \frac{1}{L_m}\sum_{i=1}^n \sum_{j=1}^{L_{i,m}}\left(\left\langle A_{i,m}^j, Q\right\rangle\right)(A_{i,m}^j)^\top W$$

$$= \boldsymbol{B} - \frac{\eta}{M}\sum_{m=1}^M \frac{1}{L_m}\sum_{i=1}^n \sum_{j=1}^{L_{i,m}}\boldsymbol{x}_{i,m}^j(\boldsymbol{x}_{i,m}^j)^\top q_m \boldsymbol{w}_m^\top$$

$$= \boldsymbol{B} - \frac{\eta}{M}Q^\top W - \underbrace{\left[\frac{\eta}{M}\sum_{m=1}^M \frac{1}{L_m}\sum_{i=1}^n \sum_{j=1}^{L_{i,m}}\boldsymbol{x}_{i,m}^j(\boldsymbol{x}_{i,m}^j)^\top q_m \boldsymbol{w}_m^\top - \frac{\eta}{M}Q^\top W\right]}_{H_Q}. \qquad (21)$$

Above, we define $q_m \in \mathbb{R}^d$ to denote the $m$-th row of $Q$ (viewed as a column vector). Note again that since

$$\frac{1}{L_m} \sum_{i=1}^{n} \sum_{j=1}^{L_{i,m}} \boldsymbol{x}_{i,m}^j (\boldsymbol{x}_{i,m}^j)^\top \approx I_d,$$

the term $H_Q$ in equation 21 can be appropriately bounded. Note the resemblance of equation 21 to (53) in (Collins et al., 2021); the crucial difference is that we will need to lower bound $\frac{1}{m}\sigma_{\min}^2(W^*)$, instead of $\frac{1}{n}\sigma_{\min}^2(W^*)$ as in (Collins et al., 2021). Thus we should be able to carry out the rest of the analysis in a similar way to the outline in (Collins et al., 2021) and derive an analogous result to Theorem 1 in (Collins et al., 2021).

We first bound the error term $H_Q$.

**Lemma A.9.** *Let*

$$H_Q^t := \frac{\eta}{M} \sum_{m=1}^{M} \frac{1}{L_m} \sum_{i=1}^{n} \sum_{j=1}^{L_{i,m}} \boldsymbol{x}_{i,m}^j (\boldsymbol{x}_{i,m}^j)^\top q_m (\boldsymbol{w}_m^{t+1})^\top - \frac{\eta}{M}(Q^t)^\top W^{t+1}.$$

*Let $\gamma_k := \frac{20k\sqrt{d}}{c\sqrt{nL}}$ for some absolute constant c. Suppose that $0 \le \gamma_k < k$. Then, for any t, with probability at least $1 - \exp(-90d) - 2e^{-50k \log M}$,*

$$\|H_Q^t\|_2 \le \eta \gamma_k \mathrm{dist}(\boldsymbol{B}^*, \boldsymbol{B}^t).$$

*Proof.* As before, we may omit the time superscript $t$ in cases where it is clear for notational convenience. The proof is based on the argument in Lemma 5 in (Collins et al., 2021). Again, the main tool is an $\epsilon$-net argument. We first bound $\|q_m\|_2$ and $\|\boldsymbol{w}_m\|_2$.

**Bounding $q_m$:** With probability at least $1 - 2e^{-50k \log M}$, for each $m \in [M]$, we have that

$$\begin{aligned}
\|q_m\|_2 &= \left\| \boldsymbol{B}^t((\boldsymbol{B}^t)^\top \boldsymbol{B}^* \boldsymbol{w}_m^* + F_m) - \boldsymbol{B}^* \boldsymbol{w}_m^* \right\|_2 \\
&\le \left\| (\boldsymbol{B}^t(\boldsymbol{B}^t)^\top - I)\boldsymbol{B}^* \boldsymbol{w}_m^* \right\|_2 + \left\| \boldsymbol{B}^t F_m \right\|_2 \\
&\le \mathrm{dist}(\boldsymbol{B}^t, \boldsymbol{B}^*)\|\boldsymbol{w}_m^*\|_2 + \|F_m\|_2 \\
&\le \sqrt{k}\mathrm{dist}(\boldsymbol{B}^t, \boldsymbol{B}^*) + \frac{\delta_k}{1 - \delta_k}\mathrm{dist}(\boldsymbol{B}^t, \boldsymbol{B}^*)\|\boldsymbol{w}_m^*\|_2 \\
&\le 2\sqrt{k}\mathrm{dist}(\boldsymbol{B}^t, \boldsymbol{B}^*).
\end{aligned}$$

Above, we utilized the assumption that $\|\boldsymbol{w}_m^*\|_2 \le \sqrt{k}$, the orthonormality of $\boldsymbol{B}^t$ (which was derived as the orthogonal matrix from a Gram-Schmidt procedure), the assumption that $0 < \delta_k \le 1/2$, as well Lemma A.8 which bounds $\|F_m\|_2$ (for all $m$) with probability at least $1 - 2e^{-50k \log M}$.

**Bounding $\boldsymbol{w}_m$:** Note that for notational convenience, we let $\boldsymbol{w}_m$ denote $\boldsymbol{w}_m^{t+1}$. For each $t$ and every $m \in [M]$, we have that

$$\begin{aligned}
\|\boldsymbol{w}_m^{t+1}\|_2 &= \left\| (\boldsymbol{B}^t)^\top \boldsymbol{B}^* \boldsymbol{w}_m^* + F_m \right\|_2 \\
&\le \|\boldsymbol{w}_m^*\|_2 + \|F_m\|_2 \\
&\le \|\boldsymbol{w}_m^*\|_2 + \frac{\delta_k}{1 - \delta_k}\mathrm{dist}(\boldsymbol{B}^t, \boldsymbol{B}^*)\|\boldsymbol{w}_m^*\|_2 \\
&\le 3\sqrt{k},
\end{aligned}$$

with probability at least $1 - 2e^{-50k \log M}$, where again we used Lemma A.8 to handle $\|F_m\|_2$, the assumption that $\delta_k < 1/2$, and the fact that $\mathrm{dist}(\boldsymbol{B}^t, \boldsymbol{B}^*) \le 2$.

For the rest of the proof, we condition on the event

$$\mathcal{E} := \left\{ \|q_m\|_2 \le 2\sqrt{k}\mathrm{dist}(\boldsymbol{B}^t, \boldsymbol{B}^*) \text{ and } \|\boldsymbol{w}_m\|_2 \le 3\sqrt{k} \quad \forall m \in [M] \right\},$$

which holds with probability at least $1 - 2e^{-50k \log M}$.

$\epsilon$-**net argument to bound** $H_Q$**:** Again, note that there exists an $1/4$-net $\mathcal{N}_k$ of the unit sphere $S^{k-1}$ and an $1/4$-net $\mathcal{N}_d$ of the unit sphere $S^{d-1}$ with cardinalities less than or equal to $9^k$ and $9^d$ respectively.

Note now that by Equation 4.13 in (Vershynin, 2018), we have

$$
\begin{aligned}
\|H_Q\|_2 &= \left\| \frac{\eta}{M} \sum_{m=1}^{M} \frac{1}{L_m} \sum_{i=1}^{n} \sum_{j=1}^{L_{i,m}} \boldsymbol{x}_{i,m}^j (\boldsymbol{x}_{i,m}^j)^\top q_m \boldsymbol{w}_m^\top - \frac{\eta}{M} Q^\top W \right\|_2 \\
&\leq 2\eta \max_{u \in \mathcal{N}_d, v \in \mathcal{N}_k} \frac{1}{M} \sum_{m=1}^{M} \frac{1}{L_m} \sum_{i=1}^{n} \sum_{j=1}^{L_{i,m}} \left\langle \left( \boldsymbol{x}_{i,m}^j (\boldsymbol{x}_{i,m}^j)^\top q_m \boldsymbol{w}_m^\top - q_m \boldsymbol{w}_m^\top \right) u, v \right\rangle \\
&= 2\eta \max_{u \in \mathcal{N}_d, v \in \mathcal{N}_k} \frac{1}{M} \sum_{m=1}^{M} \frac{1}{L_m} \sum_{i=1}^{n} \sum_{j=1}^{L_{i,m}} \left[ \left( u^\top \boldsymbol{x}_{i,m}^j \right) \left( (\boldsymbol{x}_{i,m}^j)^\top q_m \boldsymbol{w}_m^\top v \right) - \langle q_m \boldsymbol{w}_m^\top u, v \rangle \right]
\end{aligned}
$$

(22)

Fix now a $u \in \mathcal{N}_d$ and $v \in \mathcal{N}_k$. Note now that $\left( u^\top \boldsymbol{x}_{i,m}^j \right) \left( (\boldsymbol{x}_{i,m}^j)^\top q_m \boldsymbol{w}_m^\top v \right)$ is subexponential with norm less than or equal to $\|q_m\|_2 \|\boldsymbol{w}_m\|_2 \leq 6k \mathrm{dist}(\boldsymbol{B}^t, \boldsymbol{B}^*)$, since it is the product of two subgaussian variables $u^\top \boldsymbol{x}_{i,m}^j$ and $(\boldsymbol{x}_{i,m}^j)^\top q_m \boldsymbol{w}_m^\top v$ with subgaussian norms bounded by 1 and $\|q_m\|_2 \|\boldsymbol{w}_m\|_2$ respectively. Note also that

$$
\mathbb{E}\left[ \left( u^\top \boldsymbol{x}_{i,m}^j \right) \left( (\boldsymbol{x}_{i,m}^j)^\top q_m \boldsymbol{w}_m^\top \right) \right] = \mathbb{E}\left[ \langle q_m \boldsymbol{w}_m^\top u, v \rangle \right].
$$

Thus, by Bernstein's inequality, carrying on from equation 22, we have that

$$
\begin{aligned}
&\mathbb{P}\left( \frac{1}{M} \sum_{m=1}^{M} \frac{1}{L_m} \sum_{i=1}^{n} \sum_{j=1}^{L_{i,m}} \left[ \left( u^\top \boldsymbol{x}_{i,m}^j \right) \left( (\boldsymbol{x}_{i,m}^j)^\top q_m \boldsymbol{w}_m^\top v \right) - \langle q_m \boldsymbol{w}_m^\top u, v \rangle \right] \geq \rho \right) \\
&\leq \exp\left( -cnL \min\left( \frac{\rho}{6k \mathrm{dist}(\boldsymbol{B}^t, \boldsymbol{B}^*)}, \left( \frac{\rho}{k \mathrm{dist}(\boldsymbol{B}^t, \boldsymbol{B}^*)} \right)^2 \right) \right) \\
&\leq \exp\left( -cnL \left( \frac{\rho}{k \mathrm{dist}(\boldsymbol{B}^t, \boldsymbol{B}^*)} \right)^2 \right),
\end{aligned}
$$

where we will choose $\rho$ such that $\frac{\rho}{k \mathrm{dist}(\boldsymbol{B}^t, \boldsymbol{B}^*)} \leq 1$ to simplify the exponent in the way we did, and $c$ is an absolute constant that may change from line to line. Above, we also used the fact that $\sum_{m=1}^{M} L_m = nL$ (recall that $L$ is the total number of samples per agent and there are $n$ agents).

Consider the choice

$$
\rho = 10 \frac{k \sqrt{d} \mathrm{dist}(\boldsymbol{B}^t, \boldsymbol{B}^*)}{c \sqrt{nL}}.
$$

Then,

$$
\begin{aligned}
&\mathbb{P}\left( \frac{1}{M} \sum_{m=1}^{M} \frac{1}{L_m} \sum_{i=1}^{n} \sum_{j=1}^{L_{i,m}} \left[ \left( u^\top \boldsymbol{x}_{i,m}^j \right) \left( (\boldsymbol{x}_{i,m}^j)^\top q_m \boldsymbol{w}_m^\top v \right) - \langle q_m \boldsymbol{w}_m^\top u, v \rangle \right] \geq \rho \right) \\
&\leq \exp\left( -cnL \left( \frac{\rho}{k \mathrm{dist}(\boldsymbol{B}^t, \boldsymbol{B}^*)} \right)^2 \right) \\
&\leq \exp(-100d).
\end{aligned}
$$

Taking a union bound over all $u \in \mathcal{N}_d$ and $v \in \mathcal{N}_k$, it follows then that

$$
\mathbb{P}\left( \frac{\|H_Q\|_2}{\eta} \geq 2\rho \right) \leq 9^{d+k} \exp(-100d) \leq \exp(-90d),
$$

where above we used the fact that $d \geq k$. $\qquad\square$

### A.4.4 COMBINING EARLIER ARGUMENT: CONVERGENCE OF FEDDAR

As seen in Lemma A.8, we require that $L_{\min} := \min_{m \in [M]} L_m$ to be lower bounded. However, since $L_m$ is a stochastic variable, we are unable to directly lower bound it. Below, we provide a result that converts a lower bound on each client's sample size $L$ (a deterministic quantity we can control) to a high-probability lower bound on $L_{\min}$.

**Lemma A.10.** *Let $L_{\min} := \min_{m \in [M]} L_m$. For any $\alpha > 0$, suppose that for each $m \in [M]$,*

$$L \geq \max \left\{ \frac{182 \log M}{\sum_{i=1}^n \pi_{i,m}}, \frac{16}{\sum_{i=1}^n \pi_{i,m}}, \frac{2\alpha}{\sum_{i=1}^n \pi_{i,m}} \right\}.$$

*Then, with probability at least $1 - \exp(-90)$,*

$$L_{\min} \geq \alpha.$$

*Proof.* Note that

$$L_m = \sum_{i=1}^n \sum_{j=1}^L 1(\text{domain}(x_i^j) = m),$$

which is a sum of $nL$ independent random variables bounded between 0 and 1. Moreover,

$$\mathbb{E}[L_m] = \sum_{i=1}^n \pi_{i,m} L,$$

where $\pi_{i,m}$ is the probability that a datapoint comes from domain $m$ for client $i$. Note finally that

$$\mathbb{E}\left[ \left( 1(\text{domain}(x_i^j) = m) \right)^2 \right] = \pi_{i,m}.$$

Hence, by Bernstein's inequality, it follows that for any $s > 0$,

$$\mathbb{P}\left( L_{i,m} \leq \sum_{i=1}^n \pi_{i,m} L - s \right) \leq \exp\left( -\frac{s^2/2}{\sum_{i=1}^n \sum_{j=1}^L \pi_{i,m} + s/3} \right).$$

Since we wish to perform union bound over the $M$ domains, we seek to choose $s$ and $L$ such that

$$\exp\left( -\frac{s^2/2}{\sum_{i=1}^n \sum_{j=1}^L \pi_{i,m} + s/3} \right) \leq \exp\left( -91 \log M \right),$$

so that

$$M \exp\left( -\frac{s^2/2}{\sum_{i=1}^n \sum_{j=1}^L \pi_{i,m} + s/3} \right) \leq M \exp\left( -91 \log M \right) \leq \exp\left( -90 \log M \right).$$

To this end, note that we need

$$\frac{s^2/2}{\sum_{i=1}^n \sum_{j=1}^L \pi_{i,m} + s/3} \geq 91 \log M$$

$$\iff s^2 \geq 2 \cdot 91 \log M \left( \sum_{i=1}^n \sum_{j=1}^L \pi_{i,m} + s/3 \right)$$

$$\iff s \geq \sqrt{182 \log M \sum_{i=1}^n \pi_{i,m} L + \left( \frac{182 \log M}{3} \right)^2} + \frac{182 \log M}{3}$$

Suppose we pick $L$ such that

$$\sum_{i=1}^n \pi_{i,m} L \geq 182 \log M,$$

so that

$$\sqrt{182 \log M \sum_{i=1}^{n} \pi_{i,m} L + \left(\frac{182 \log M}{3}\right)^2} + \frac{182 \log M}{3} \le 2\sqrt{\sum_{i=1}^{n} \pi_{i,m} L}.$$

Then, by picking $s = 2\sqrt{\sum_{i=1}^{n} \pi_{i,m} L}$, it follows that

$$\exp\left(-\frac{s^2/2}{\sum_{i=1}^{n} \sum_{j=1}^{L} \pi_{i,m} + s/3}\right) \le \exp\left(-91 \log M\right),$$

such that for each $m \in [M]$,

$$\mathbb{P}\left(L_{i,m} \le \sum_{i=1}^{n} \pi_{i,m} L - 2\sqrt{\sum_{i=1}^{n} \pi_{i,m} L}\right) \le \exp(-91 \log M).$$

By choosing $L$ such that

$$\sqrt{\sum_{i=1}^{n} \pi_{i,m} L} \ge 4,$$

it follows that

$$\mathbb{P}\left(L_{i,m} \le \frac{\sum_{i=1}^{n} \pi_{i,m} L}{2}\right) \le \exp(-91 \log M).$$

The result now follows by choosing $L$ such that it also satisfies

$$\frac{\sum_{i=1}^{n} \pi_{i,m} L}{2} \ge \alpha$$

for each $m$.

$\square$

**Lemma A.11** (Descent lemma). *Define $E_0 := 1 - \mathrm{dist}^2(\boldsymbol{B}^0, \boldsymbol{B}^*)$ and $\bar{\sigma}_{\max,*} := \sigma_{\max}\left(\frac{1}{\sqrt{M}} W^*\right)$ and $\bar{\sigma}_{\min,*} := \sigma_{\min}\left(\frac{1}{\sqrt{M}} W^*\right)$. Let $\kappa := \frac{\bar{\sigma}_{\max,*}}{\bar{\sigma}_{\min,*}}$. Consider any iteration $t$.*

*Suppose that*

$$L \ge \left(\frac{400 dk^2}{nc}\right) \frac{1}{\left(\min\left\{\frac{1}{2}, 8E_0/(25 \cdot 5\kappa^2)\right\}\right)^2}, \tag{23}$$

*where $c > 0$ is absolute constant. Suppose also that*

$$L \ge \max\left\{\frac{182 \log M}{\sum_{i=1}^{n} \pi_{i,m}}, \frac{16}{\sum_{i=1}^{n} \pi_{i,m}}, \frac{2\left(100 Ck \log M\right)\frac{1}{\left(\min\left\{\frac{1}{2}, 8E_0/(25 \cdot 5\kappa^2)\right\}\right)^2}}{\sum_{i=1}^{n} \pi_{i,m}}\right\},$$

*which by Lemma A.10, ensures that with probability at least $1 - e^{-90}$,*

$$L_{\min}^{t} \ge \left(100 Ck \log M\right) \frac{1}{\left(\min\left\{\frac{1}{2}, 8E_0/(25 \cdot 5\kappa^2)\right\}\right)^2}, \tag{24}$$

*where $L_{\min}^{t} = \min_{m \in [M]} L_m^t$ denotes the minimum number of samples from any domain at iteration $t$, and $C > 0$ is an absolute constant.*

*Then, for any $\eta \le 1/(4\bar{\sigma}_{\max,*}^2)$, we have*

$$\mathrm{dist}(\boldsymbol{B}^{t+1}, \boldsymbol{B}^*) \le (1 - \eta E_0 \bar{\sigma}_{\min,*}/2)^{1/2} \mathrm{dist}(\boldsymbol{B}^t, \boldsymbol{B}^*),$$

*with probability at least $1 - e^{-80}$.*

*Proof.* We begin with the observation that

$$W^{t+1} = W^*(\boldsymbol{B}^*)^\top \boldsymbol{B}^t + F^t$$
$$\bar{\boldsymbol{B}}^{t+1} = \boldsymbol{B}^t - \frac{\eta}{M}(Q^t)^\top W^{t+1} - H_Q^t,$$

where

$$Q^t = W^{t+1}(\boldsymbol{B}^t)^\top - (W^*)(\boldsymbol{B}^*)^\top,$$

and

$$H_Q^t := \frac{\eta}{M}\sum_{m=1}^M \frac{1}{L_m}\sum_{i=1}^n \sum_{j=1}^{L_{i,m}} \boldsymbol{x}_{i,m}^j (\boldsymbol{x}_{i,m}^j)^\top q_m (\boldsymbol{w}_m^{t+1})^\top - \frac{\eta}{M}(Q^t)^\top W^{t+1}.$$

Above $\bar{\boldsymbol{B}}^{t+1}$ denotes the estimate of $\boldsymbol{B}$ before we perform the $QR$ decomposition. We note that the updates for $W$ and $\boldsymbol{B}$ are exactly analogous to the updates for $W$ and $\boldsymbol{B}$ as seen in the proof of Lemma 6 in (Collins et al., 2021). The only two differences are

1. The definitions of $F$ in our paper and (Collins et al., 2021) are slightly different. However, in both cases,

$$\|F\|_F \le \frac{\delta_k}{1-\delta_k}\mathrm{dist}(\boldsymbol{B}^*, \boldsymbol{B})\|W^*\|_F$$

   for some term $\delta_k \le 1/2$ with high probabilities. In our case, this event holds with probability at least $1 - 2e^{-50k \log M}$, whilst in (Collins et al., 2021), the event holds with probability at least $1 - \exp(-110k^2 \log n)$.

2. The update for $\bar{\boldsymbol{B}}^{t+1}$ in (Collins et al., 2021) takes the form

$$\bar{\boldsymbol{B}}^{t+1} = \boldsymbol{B}^t - \frac{\eta}{rn}(Q^t)^\top W^{t+1} - \frac{\eta}{rn}\left(\frac{1}{m}\mathcal{A}^\dagger \mathcal{A}(Q^t) - Q^t\right)^\top W^{t+1},$$

   where $0 \le r \le 1$ is a ratio term used in (Collins et al., 2021), and $m$ above represents the number of samples used by each learner in (Collins et al., 2021) (which is different from our use of $m$ as an index over the domains). However, we note that with high probabilities,

$$\left\|H_Q^t\right\|_2 \le \eta\gamma_k \mathrm{dist}(\boldsymbol{B}^t, \boldsymbol{B}^*),$$
$$\left\|\frac{\eta}{rn}\left(\frac{1}{m}\mathcal{A}^\dagger \mathcal{A}(Q^t) - Q^t\right)^\top W^{t+1}\right\|_2 \le \eta\gamma_k \mathrm{dist}(\boldsymbol{B}^t, \boldsymbol{B}^*),$$

   where the definition of $\gamma_k$ in both papers differ but both satisfy the assumption that $\gamma_k \le k$.

Due to these similarities in the updates for $W^{t+1}$ and $B^{t+1}$ with the update in (Collins et al., 2021), the proof of this lemma follows naturally from the proof of Lemma 6 in (Collins et al., 2021), by plugging in $\frac{\eta}{M}(Q^t)^\top W^{t+1}$ in the update for $\bar{\boldsymbol{B}}^{t+1}$ in place of $\frac{\eta}{rn}(Q^t)^\top W^{t+1}$ as in (Collins et al., 2021). In particular, following the same analysis as in (Collins et al., 2021), we see that on the events in Lemma A.8 and Lemma A.9, following the equation immediately after Equation (84) in (Collins et al., 2021), we have

$$\mathrm{dist}(\boldsymbol{B}^t, \boldsymbol{B}^*) \le \frac{1}{\sqrt{1 - 4\eta\frac{\bar{\delta}_k}{(1-\bar{\delta}_k)^2}\bar{\sigma}_{\mathrm{max},*}{}^2}}\left(1 - \eta\bar{\sigma}_{\mathrm{min},*}^2 E_0 + 2\eta\frac{\bar{\delta}_k}{(1-\bar{\delta}_k)^2}\bar{\sigma}_{\mathrm{max},*}^2\right)\mathrm{dist}(\boldsymbol{B}^t, \boldsymbol{B}^*),$$

where in our case $\bar{\delta}_k = \delta_k + \gamma_k$. Then, by choosing

$$\bar{\delta}_k < 16E_0/(25 \cdot 5\kappa^2), \tag{25}$$

it follows that $\bar{\delta}_k < 1/5$, and so

$$1 - \eta\bar{\sigma}_{\mathrm{min},*}^2 E_0 + 2\eta\frac{\bar{\delta}_k}{(1-\bar{\delta}_k)^2}\bar{\sigma}_{\mathrm{max},*}^2 \le 1 - 4\eta\frac{\bar{\delta}_k}{(1-\bar{\delta}_k^2)}\bar{\sigma}_{\mathrm{max},*}^2 \le 1 - \eta E_0\bar{\sigma}_{\mathrm{min},*}^2/2,$$

as in equation (85) in (Collins et al., 2021), such that

$$\text{dist}(\boldsymbol{B}^{t+1}, \boldsymbol{B}^*) \leq (1 - \eta E_0 \bar{\sigma}_{\min,*}^2/2)^{1/2} \text{dist}(\boldsymbol{B}^t, \boldsymbol{B}^*).$$

It remains for us to understand what the constraint on $\bar{\bar{\delta}}_k$ spelt out in equation 25, and the constraints on $\delta_k$ and $\gamma_k$ (in Lemmas A.8 and A.9 respectively) mean in our choice of the sample size $L$ for each agent, and the domain size $L_m$ at each iteration. Observe that we need

$$\delta_k = \frac{10C\sqrt{k}\sqrt{\log M}}{\sqrt{L_{\min}}} \leq \frac{1}{2}, \tag{26}$$

$$\gamma_k = \frac{20k\sqrt{d}}{c\sqrt{nL}} \leq \frac{1}{2}, \tag{27}$$

$$\bar{\delta}_k = \delta_k + \gamma_k = \frac{10C\sqrt{k}\sqrt{\log M}}{\sqrt{L_{\min}}} + \frac{20k\sqrt{d}}{c\sqrt{nL}} \leq 16E_0/(25 \cdot 5\kappa^2), \tag{28}$$

where $c, C > 0$ are absolute constants. By choosing

$$L_{\min} \geq (100Ck \log M) \frac{1}{\left(\min\left\{\frac{1}{2}, 8E_0/(25 \cdot 5\kappa^2)\right\}\right)^2}$$

$$L \geq \left(\frac{400dk^2}{nc}\right) \frac{1}{\left(\min\left\{\frac{1}{2}, 8E_0/(25 \cdot 5\kappa^2)\right\}\right)^2},$$

we ensure that the requirements in equation 26, equation 27 and equation 28 are all satisfied.

The final result then follows by applying Lemma A.10. □

This then yields the following convergence result, which is a more complete statement of A.4.

**Theorem A.12** (Convergence result for Algorithm 2). *Define $E_0 := 1 - \text{dist}^2(\boldsymbol{B}^0, \boldsymbol{B}^*)$ and $\bar{\sigma}_{\max,*} := \sigma_{\max}\left(\frac{1}{\sqrt{M}}W^*\right)$ and $\bar{\sigma}_{\min,*} := \sigma_{\min}\left(\frac{1}{\sqrt{M}}W^*\right)$. Let $\kappa := \frac{\bar{\sigma}_{\max,*}}{\bar{\sigma}_{\min,*}}$.*

*Suppose that*

$$L \geq \left(\frac{400dk^2}{nc}\right) \frac{1}{\min\left\{\frac{1}{2}, 8E_0/(25 \cdot 5\kappa^2)\right\}},$$

*where $c > 0$ is absolute constant. Suppose also that*

$$L \geq \max\left\{\frac{182 \log M}{\sum_{i=1}^n \pi_{i,m}}, \frac{16}{\sum_{i=1}^n \pi_{i,m}}, \frac{2(100Ck \log M)\frac{1}{\min\{1/2, 8E_0/(25 \cdot 5\kappa^2)\}}}{\sum_{i=1}^n \pi_{i,m}}\right\}.$$

*Then, for any $\eta \leq 1/(4\bar{\sigma}_{\max,*}^2)$, we have*

$$\text{dist}(\boldsymbol{B}^{t+1}, \boldsymbol{B}^*) \leq (1 - \eta E_0 \bar{\sigma}_{\min,*}/2)^{1/2} \text{dist}(\boldsymbol{B}^t, \boldsymbol{B}^*),$$

*with probability at least $1 - e^{-80}$. Then for any $T$ and any $\eta \leq 1/(4\sigma_{\max,*}^2)$, we have*

$$\text{dist}(\boldsymbol{B}^t, \boldsymbol{B}^*) \leq (1 - \eta E_0 \bar{\sigma}_{\min,*}^2/2)^{T/2} \text{dist}(\boldsymbol{B}^0, \boldsymbol{B}^*), \tag{29}$$

*with probability at least $1 - Te^{-80}$.*

By assuming that $\sigma_{\min,*}^2 > 0$, the bound in Theorem 1 decays exponentially. We note that the total number of samples required per client scales with $L \log(1/\epsilon)$. In addition, in order for the result to be meaningful, we implicitly assume that $E_0$ is close to 1 such that

$$0 < 1 - \eta E_0 \bar{\sigma}_{\min}^2 < 1.$$

To do so, we note it is possible to choose $\boldsymbol{B}^0$ such that $\text{dist}(\boldsymbol{B}_0, \boldsymbol{B}^*)$ is close enough to 0, with only a logarithmic increase in sample complexity when the number of samples is uniform across the domains. The argument follows the proof of Theorem 3 in (Tripuraneni et al., 2021).

**Theorem A.13.** *Suppose Assumptions A.1, A.2, A.3 all hold. Suppose also that $x_i^{0,j} \sim \mathcal{N}(0, I_d)$ independently for all $i \in [n]$. Suppose each client $i$ sends the server $Z_i := \sum_{j=1}^{L^0} (y_i^{0,j})^2 x_i^j (x_i^j)^\top$, as well as the integer value of $L_i$, such that the server can compute $Z := \frac{1}{nL^0} \sum_{i=1}^n Z_i$. Then, the server computes $UDU^\top \leftarrow$ rank-k SVD $(Z)$, and sets $\boldsymbol{B}^0 := U$. Let*

$$\bar{\Lambda} = \frac{1}{nL^0} \sum_{i=1}^n \sum_{j=1}^{L^0} w_{m(i,j)}^* (w_{m(i,j)}^*)^\top,$$

*where $m(i, j)$ denotes the sample of the $j$-th sample from the $i$-th client. Let $\sigma_{\min,*} := \sigma_{\min}(\bar{\Lambda})$, and let $\sigma_{\max,*} := \sigma_{\max}(\bar{\Lambda})$. Suppose that $L^0 \geq c\mathrm{polylog}(d, nL^0)\sigma_{\max,*}dk^2/(n\sigma_{\min,*}^2)$. Then, with probability at least $1 - (nL^0)^{-100}$, we have that*

$$\mathrm{dist}(\boldsymbol{B}^0, \boldsymbol{B}^*)^2 \leq \tilde{O}\left(\frac{\sigma_{\max,*}k^2 d}{\sigma_{\min,*}^2 nL^0}\right).$$

*In particular, when the number of samples is uniform across the domains, we have that*

$$\mathrm{dist}(\boldsymbol{B}^0, \boldsymbol{B}^*)^2 \leq \tilde{O}\left(\frac{\kappa^4 k^2 d}{nL^0}\right),$$

*where we recall that $\kappa := \bar{\sigma}_{\max,*}/\bar{\sigma}_{\min,*}$, and*

$$\bar{\sigma}_{\max,*} := \sigma_{\max}\left(\frac{1}{\sqrt{M}}W^*\right), \quad \bar{\sigma}_{\min,*} := \sigma_{\min}\left(\frac{1}{\sqrt{M}}W^*\right).$$

*Proof.* We omit the proof since it is a slight variant of Theorem 3 in (Tripuraneni et al., 2021). For completeness, note that in the case when the number of samples is uniform across the domains, some algebra shows that

$$\mathrm{dist}(\boldsymbol{B}^0, \boldsymbol{B}^*)^2 \leq \tilde{O}\left(\frac{\kappa^2 k^2 d}{\bar{\sigma}_{\min,*}^2 nL^0}\right).$$

However, since $k/4M \leq \|W^*\|_F^2 \leq kM\bar{\sigma}_{\max,*}^2$, we have that

$$\frac{1}{\bar{\sigma}_{\min,*}^2} = \kappa^2 \frac{1}{\bar{\sigma}_{\max,*}^2} \leq 4\kappa^2,$$

which proves the last statement in the theorem.

$\square$

# B ADDITIONAL EXPERIMENTAL RESULTS

## B.1 EXPERIMENTS ON FAIRFACE DATASET FOR GENDER CLASSIFICATION

Table 4: Min, max and average test accuracy of gender classification across 7 domains (*race groups*) on FairFace with number of clients $n = 5$, number of samples at each client $L_i = 500$.

| Task | Method | $\alpha = 0.1$ | | | $\alpha = 0.5$ | | | $\alpha = 1$ | | | $\alpha = 100$ | | |
|------|--------|-----|-----|-----|-----|-----|-----|-----|-----|-----|-----|-----|-----|
| | | Max | Min | Avg | Max | Min | Avg | Max | Min | Avg | Max | Min | Avg |
| Gender | FedAvg | **92.0** | 71.7 | 83.9 | 89.8 | 77.6 | 84.5 | 91.0 | 77.4 | 84.2 | 90.5 | 77.1 | 84.7 |
| | FedAvg + Multi-head | 90.2 | 48.7 | 78.9 | 89.2 | 77.8 | 84.1 | **91.6** | 76.8 | 83.9 | 91.1 | 77.5 | 84.5 |
| | FedDAR-WA | 89.8 | 53.4 | 80.9 | **91.5** | 76.7 | 84.3 | 91.2 | 76.1 | 84.3 | 90.0 | 76.8 | 84.1 |
| | FedDAR-SA | **92.2** | **73.4** | **85.1** | 91.3 | **78.1** | **85.2** | 91.4 | **78.2** | **85.1** | **92.2** | **78.1** | **85.6** |

We also conduct experiments for gender classification on FairFace with the same settings. The best representation dimension is $k = 2$ for this task, probably due to the smaller diversity across the domains. We can see that the results shown in Table 4 have a similar trend as the results in Table 5.2.

## B.2 EXPERIMENTS ON DIGITS DATASET

Table 5: Min, max and average test accuracy of digits classification across 5 domains with number of clients $n = 5$, number of samples at each client $L_i = 500$.

| Method | $\alpha = 0.1$ | | | $\alpha = 0.5$ | | | $\alpha = 1$ | | | $\alpha = 100$ | | |
|--------|-----|-----|-----|-----|-----|-----|-----|-----|-----|-----|-----|-----|
| | Max | Min | Avg | Max | Min | Avg | Max | Min | Avg | Max | Min | Avg |
| FedAvg | **97.1** | **60.7** | **80.6** | 97.2 | 64.3 | 81.7 | 96.1 | **74.8** | 85.2 | 96.8 | **71.0** | 85.1 |
| FedAvg + Multi-head | 94.3 | 26.5 | 55.9 | 94.3 | 44.8 | 68.3 | 94.1 | 56.7 | 74.6 | 95.0 | 52.3 | 74.5 |
| FedDAR-WA | **97.3** | 52.3 | 79.8 | **97.3** | **64.7** | **83.1** | **96.6** | 74.5 | **86.3** | **97.1** | 70.6 | **86.3** |

We perform additional experiments on the digits dataset with five data domains with feature shift (Li et al., 2021b). Details are described in the following paragraphs. From Table 5, we can see that FedDAR-WA outperforms FedAvg consistently except in the case where domain distributions are extremely heterogeneous ($\alpha = 0.1$). In this case, each client tends to have data from only one domain. It is difficult for the proposed method to learn a good domain-specific head for the domain with the most different data (more obvious feature shift) under this circumstance. For other levels of heterogeneity, although the min and max domain accuracies are similar between FedAvg and FedDAR-WA, the average accuracies are improved as a result of the domain-wise personalized model. On the other hand, without an alternative update of the head and representation, FedAvg + Multi-head will overfit quickly. We don't include the results of FedDAR-SA here because using representation dimension $k \geq 64$ causes numerical instability during head aggregation and failure to converge. While using representation dimension $k \leq 32$ leads to lower accuracy.

**Datasets.** We use the same digits dataset containing five different data domains as (Li et al., 2021b). Specifically, we use SVHN (Netzer et al., 2011), USPS (Hull, 1994), SynthDigits (Ganin & Lempitsky, 2015), MNIST-M (Ganin & Lempitsky, 2015) and MNIST (LeCun et al., 1998) as five data domains. Similarity to the experiments on FairFace datraset, the training data is divided into $n$ clients without duplication. Each client has a domain distribution $\boldsymbol{\pi_i} \sim Dir(\alpha\boldsymbol{p})$ sampled from a Dirichlet distribution.

**Implementation Details.** We adapt the codebase from (Li et al., 2021b). A 6-layer CNN with 3 convolutional layers and 3 fully-connected layers is used, with the last layer as domain-specific head. We use SGD optimizer with learning rate $10^{-2}$ and cross-entropy loss. The batch size is set to 32, and the total communication rounds is set to 100. For each method, we first train the model for 10 rounds with 1 local epoch using FedAvg as warmup. The accuracy shown is the average over the last ten communication rounds. We repeat experiment for each setting three times with different random seeds and report the averages.

### B.3 Further experimental details

#### B.3.1 Synthetic Data

For the synthetic data experiments, we adapt the code from (Collins et al., 2021) and follow a similar protocol. The ground-truth matrices $\boldsymbol{W}^* \in \mathbb{R}^{M \times k}$ and $\boldsymbol{B}^* \in \mathbb{R}^{d \times k}$ are generated following the same way as (Collins et al., 2021) by sampling each element from i.i.d. standard normal distribution and taking the QR factorization. The same $L$ samples are used for each client during the whole training process. Test samples are generated in the same way as the traning samples but without noise. For all the methods, models are initalized with ramdom Gaussian samples. We set $\alpha = 0.4$ for experiments in Figure 5.2.

#### B.3.2 Real data with controlled distribution

**Implementation details.**   We use Imagenet(Deng et al., 2009) pre-trained ResNet-34 (He et al., 2016) for all experiments on this dataset. All the methods are trained for $T = 100$ communication rounds, with 20 rounds of FedAvg as warmup. For FedDAR-WA and FedDAR-SA, 5 epochs of local updates are executed for both heads and representation at each round. For the baselines, 5 epochs of local updates are executed at each round for fair comparison. We use Adam optimizer with a learning rate of $1 \times 10^{-4}$ for the first 60 rounds and $1 \times 10^{-5}$ for the last 40 rounds. The images are resized to $224 \times 224$ with only random horizontal flip for augmentation. The learning rate and the number of local epochs is tuned by grid search with a fixed batch size of 64. We tuned the projection dimension $k$ for FedDAR-SA among {4,8,16,32,64} with $\alpha = 1.0$ and used $k = 8$ for all other $\alpha$.

Our evaluation metrics are the classification accuracy on the whole validation set of FairFace for each race group. We don't have extra local validation set to each client since we assume the data distribution within each domain is consistent across the clients. The numbers reported are the average over the final 10 rounds of communication following the standard practice in (Collins et al., 2021), and the average of three independent runs with different random seeds.

#### B.3.3 Real data with real-World data distribution

**Dataset details.**   The detailed statistics of the partial EXAM dataset is summarized in Table 6. The "Other" category includes American Indian or Alaska native, native Hawaiian or other Pacific islander and patients with more than one race or unknown race. $\geq$HFO % means the percentage of cases with positive labels (receiving oxygen therapy higher or equal to high-flow oxygen with 72 hours).

Table 6: Data summary of the partial EXAM dataset used in our study.

| Site | White | Black | Asian | Latino | Other | $\geq$HFO % |
|---|---|---|---|---|---|---|
| Site-1 | 59.6% | 10.0% | 3.4% | 2.0% | 24.9% | 12.4% |
| Site-2 | 75.0% | 11.1% | 2.8% | 0.6% | 10.5% | 9.1% |
| Site-3 | 46.5% | 26.3% | 4.2% | 7.0% | 16.0% | 9.6% |
| Site-4 | 71.4% | 6.3% | 4.2% | 0.8% | 17.2% | 11.4% |
| Site-5 | 44.0% | 28.4% | 1.6% | 6.3% | 19.8% | 9.9% |
| Site-6 | 0.0% | 0.0% | 100.0% | 0.0% | 0.0% | 18.8% |

**Implementation details.**   We apply 5-fold cross validation. The input of the model is one chest x-ray image resized to 224x224 paired with a 22-dimensional electronic health record(EHR) data, the representation dimension is 278 if it is not projected. All the models are trained for $T = 20$ communication rounds with Adam optimizer and a learning rate of $1 \times 10^{-4}$. For each round we do 1 local epoch for all the methods. For all the methods, the models are initialized with the same pretrained model as in (Dayan et al., 2021) without any warmup. For FedDAR-SA and FedDAR-WA, we execute 5 epochs of update for heads on each round, and set representation dimension $k = 16$ for FedDAR-SA. Hyperparameters including learning rate, number of epochs for head update and representation dimension are tuned through grid search with a fixed batch size of 36. For FedRep,FedDARand FedPer. For LG-FedAvg, we treated the last fully-connected layer as the global parameters and all other layers as local representation. For FedMinMax, multiple local iterations are executed during each round instead of one step of GD for reasonable comparison. For FedProx

we tuned $\mu$ among $\{0.05, 0.1, 0.25, 0.5\}$ and used $\mu = 0.1$. For the fine-tuning methods, we only fine-tune the global trained model locally with Adam optimizer and learning rate of $5e - 5$ for 1 epoch since more epochs of fine-tuning leads to worse results.

The models are evaluated by aggregating predictions on the local validation sets then calculating the area under curve (AUC) for each domain. The average AUCs on local validation set of clients are also reported. The AUC shown is first averaged over the last five communication rounds, and then averaged over five runs of 5-fold cross validation.

## C  DISCUSSION ON COMMUNICATION AND PRIVACY

**Communication**  For FedDAR-WA, the only communication overhead comes from the extra parameters of multiple heads for different domains, which only slightly increase the communication cost. For FedDAR-SA, we need to send a Hessian with $k^2 \times N^2$ parameters from each client to the server at each round. This might be costly when both representation dimension $k$ and output dimension $N$ are large. However, compared to sending millions of parameters of neural network, the extra communication cost is acceptable.

**Privacy**  For the FedDAR-WA, there is no extra parameters shared compared to FedAvg. So there is no additional privacy risk introduced. Privacy techniques like homomorphic encryption (Cheon et al., 2017) or differential privacy (McMahan et al., 2017b; Kairouz et al., 2021) that apply to FedAvg also works for FedDAR-WA. In fact, the multi-head design of different domains makes it harder to perform gradient based attack(Zhu et al., 2019) targeting our method. Because the attacker need to first figure out which domain the sample comes from. For FedDAR-SA, the only extra parameters shared is the Hessian matrices, which are aggregated results from all the local data. Recovering the information for a specific sample from Hessian is extremely difficult. Under the worst circumstance, what the attacker can recover from the Hessian is the label and the features at last layer, which hardly ease the difficulty of recovering original input.

