# OpenReview forum: "FedDAR: Federated Domain-Aware Representation Learning"
_ICLR.cc/2023/Conference — ICLR 2023 poster_

### Official Review · Reviewer_p1BK · 2022-10-23

**Confidence:** 3
**Correctness:** 3
**Technical Novelty And Significance:** 3
**Empirical Novelty And Significance:** 3
**Recommendation:** 8

**Clarity, Quality, Novelty And Reproducibility:**

# Clarity

- The paper is well written overall. There are minor typos (see below)
- I did not understand well Section 4.3. I think that some rewriting could help
  - After initially reading this section, I understood that FedDAR-SA did 1 (full-batch) local Newton step which was then re-aggregated via Eq (8), which made sense.
  - However, App B.3.3 indicates that 5 epochs of update for heads are done for FedDAR-SA. This would tend to indicate that, in fact, what is done in FedDAR-SA is a minibatch optimization of the 2nd order Taylor expansion of the loss (7). But in this case, is the hessian computed at each minibatch? And how is the aggregation done after 5 such epochs?
  - Similarly, the projection layers hinted at the last of this section are not very clear to me, compared to the rest of the paper which is very precise.

## Minor typos/comments
- Notations for $\hat{R}_{i, m}$ are not consistent in Algorithm 1 and in Eq (6) (comma vs composition operator $\circ$ )
- In algorithm 1, for the update of the trunk $\phi$, shouldn't the operator GRD apply to $\hat{R_{i}} (\phi_i)$ rather than $\hat{R_{i, m}}$?
- $\alpha$ is used both to denote the learning rate in Algorithm 1 as well as to control the Dirichlet mixtures in Sec 5, which can be confusing

# Quality

- The experiments are overall sound. My main comment is related to the choice of baselines, cf weaknesses above. I also have more minor comments:
- Regarding measurement of AUC, it is written in the last sentence of Appendix B.3.3 that AUCs are averaged over the last five communication rounds. What justifies this choice?
- Why are only cross-validation results reported for the EXAM dataset, and not a fixed train/test split?
- Would it be possible to provide error bars for the synthetic results as well as the ones with FairFace?

# Novelty

To the best of my knowledge, this is the first time that specific heads have been proposed with this formulation.

# Reproducibility

- How were parameters tuned?

**Details Of Ethics Concerns:**

No concern

**Strength And Weaknesses:**

# Strengths

- The method proposed by the authors reaches the best performance on all datasets
- Well written overall

# Weaknesses

- The main weakness of the paper stems from the way baselines are handled
  - Harmonized baselines across examples (for synthetic and fairface, separate FedAvg is not proposed although it's a natural baseline)
  - Unless mistaken, FedAvg with per-domain reweighting is not considered as a baseline, except in the top row of Table 2 (ablation study). I think it would be fairer to compare all methods with the same optimization problem, to really demonstrate the benefit of sharing the trunk in the representation (given the results of Table 2, FedDAR would still probably be the best method).
- Writing could be improved in some sections (cf detailed comments below)
- The approach is limited to known domain mixtures (acknowledged by the authors)

**Summary Of The Paper:**

This paper investigates the problem of data heterogeneity in cross-silo federated learning. It assumes that datasets in each client are composed of a mixture of known domains, and domain metadata is available for each sample.

Building on a fair learning formulation across domains (where each domain plays an equal role), the authors propose to train a neural network composed of two parts: a representation shared across domains and linear heads specific for each domain, called, FedDAR. Throughout each round, each part is trained alternatively, with the other part freezed. For the local heads, the authors consider two variants, FedDAR-WA (equivalent to FedAvg) and FedDAR-SA (second-order, requires to transmit hessians).

The paper proves on a toy linear model that the shared trunk converges to the true one linearly (Theorem 1).

Numerical experiments on synthetic data, real data with controlled distribution (fairface), and real data without controlled distribution (EXAM healthcare dataset) demonstrate that the method improves over previous personalization methods. An ablation study helps to understand the role of the different components.

**Summary Of The Review:**

Although I have some concerns regarding the choice of baselines, I think the paper makes an overall compelling case for the proposed approach. I am willing to upgrade my score if the authors address my questions.

---

> ### Author Response · Authors · 2022-11-18
> **Response to Reviewer p1BK**
>
> Thank you for reading the paper carefully and providing the constructive feedback! For the experiments part, We have added more baselines with error bars for the synthetic and FairFace results. We also revised the unclear parts, notations and typos in the paper you suggested. We address the reviewer's other comments as follows:
>
> ---*"Harmonized baselines across examples (for synthetic and fairface, separate FedAvg is not proposed although it's a natural baseline)"*
>
> We have added separate FedAvg as baselines for both synthetic and fairface in the revision.
>
> ---*"Unless mistaken, FedAvg with per-domain reweighting is not considered as a baseline, except in the top row of Table 2 (ablation study). I think it would be fairer to compare all methods with the same optimization problem, to really demonstrate the benefit of sharing the trunk in the representation (given the results of Table 2, FedDAR would still probably be the best method)."*
>
> All the original results(including FedAvg and FedAvg + Multi-head) in Table 1 are actually with per-domain reweighting for a fairer comparison as you suggested. We apologize that it was not clearly mentioned in our original submission. We fix it in our revision. We also added the results of FedAvg without per-domain reweighting for Fairface. For the synthetic data, the difference between using reweighting or not is neglectable since with client number $C=100$, $u_m=\frac{L}{L_mM}\approx 1$ for all domains. Please check the updated Figure 1 and Table 1 for new results. With highly heterogeneous domain distribution, FedAvg with per-domain reweighting has a smaller gap between maximal and minimal accuracy across domains compared to vanilla FedAvg.
>
> --*"I did not understand well Section 4.3. I think that some rewriting could help
> After initially reading this section, I understood that FedDAR-SA did 1 (full-batch) local Newton step which was then re-aggregated via Eq (8), which made sense. However, App B.3.3 indicates that 5 epochs of update for heads are done for FedDAR-SA. This would tend to indicate that, in fact, what is done in FedDAR-SA is a minibatch optimization of the 2nd order Taylor expansion of the loss (7). But in this case, is the hessian computed at each minibatch? And how is the aggregation done after 5 such epochs?"*
>
> For FedDAR-SA, We didn't actually do the (full-batch) local Newton step. As you mentioned, we did minibatch optimization of the 2nd-order Taylor expansion of the loss (7) locally for several epochs. Then we did an extra forward run on all local data to gather all the numbers (which are different depending on the loss we actually used) we need for Hessian. Then the Hessians and parameters of the heads are sent to the server for re-aggregated via Eq (8). We have rewritten this part in the revision.
>
> ---*"Similarly, the projection layers hinted at in the last of this section are not very clear to me, compared to the rest of the paper which is very precise."*
>
> The projection layer is simply one fully-connected layer added on top of the original presentation trunk. Then the presentation is projected to a lower dimensional space. We did this since the representation trunk we used was a pretrained ResNet-34. You can also directly set the embedding space to be lower dimensional if it is possible. We have updated the text to make it clearer.
>
> ---*"Regarding the measurement of AUC, it is written in the last sentence of Appendix B.3.3 that AUCs are averaged over the last five communication rounds. What justifies this choice?"*
>
> We follow the standard practice of [Collins et al., 2021] to average the metrics over the final rounds of communication. The number of rounds is chosen to be 5 based on the observation from the training curve, where we found that the training loss would plateau over the 5 last communication rounds.
>
> [Collins et al., 2021] Collins, Liam, et al. "Exploiting shared representations for personalized federated learning." International Conference on Machine Learning. PMLR, 2021.
>
> ---*"Why are only cross-validation results reported for the EXAM dataset, and not a fixed train/test split?"*
>
> The major reason we report results on 5-fold cross-validation is the limited number of positive cases, especially for those under-represented ethnic groups. From the experiments, we observe that the performance of some baselines is very sensitive to different data splits. We want to make sure that the variation coming from different data splits is shown correctly in the results.
>
> ---*"Would it be possible to provide error bars for the synthetic results as well as the ones with FairFace?"*
>
> Thanks for the suggestion. In our revision, we have included error bars in all the results (Table 1, Figure 1 and Figure 2).
>
> ---*"How were parameters tuned?"*
>
> Hyperparameters were tuned with grid search. The details have been added in Appendix B.

---

> > ### Comment · Reviewer_p1BK · 2022-12-04
> > **Thanks for your answer**
> >
> > I thank the authors for their detailed answer which addresses well my comments. I have increased my score accordingly.

---

### Official Review · Reviewer_eZTL · 2022-10-24

**Confidence:** 3
**Correctness:** 3
**Technical Novelty And Significance:** 3
**Empirical Novelty And Significance:** 3
**Recommendation:** 6

**Clarity, Quality, Novelty And Reproducibility:**

A lot of details regarding the convergence analysis are deferred to the appendix. It will be good to provide a sketch of the proof in the main paper. Please elaborate on techniques used in convergence analysis, highlighting the technical novelty of the results.

**Strength And Weaknesses:**

Strengths:

-Shared representation structure enables the algorithm to learn low-dimensional classifiers for each domain, which benefits domains with a small number of data instances.

-Experiments demonstrate that the proposed algorithm comes with significant performance gains compared with client-centric or domain-agnostic federated learning methods.

Weaknesses:

-How FedDAR optimizes the learning objective in Equation 1 is unclear. Minimizing the expected risk averaged over all domains may not lead to a model which can perform well in every domain. This paper does not address fairness issues, nor does it achieve minimization of risks on a per-client basis, as each client has a data distribution that is a different mixture of predefined domains. While it is mentioned that the algorithm uses sample re-weighting, it is not clear how this is achieved with the current objective function.

-The paper does not study the privacy implications of sharing model parameters.


**Summary Of The Paper:**

This paper considers a federated learning problem with multiple heterogeneous clients, where each client’s data distribution is a distinct mixture of predefined domain distributions. It is assumed that the domains of the data samples are known. The goal is to federatively learn a model that can perform well in every domain. For this purpose, the paper proposes FedDAR algorithm, which learns a shared representation across all domains and a separate predictor head for each domain. Training of the encoder and the head predictors are decoupled. Learning a shared representation with a small number of outputs allows domain heads to be accurately learned even with a small set of per-domain samples. Experiments show that FedDAR achieves significant performance gains over methods that do not take domain labels into account.

**Summary Of The Review:**

This work tackles the important problem of domain-aware federated learning. The proposed algorithm is intuitive, and its applicability is fortified by convergence analysis in a simplified setting and experimental results in a complex setting. While there are some issues related to privacy and computational costs, my first impression of the paper is positive.

---

> ### Author Response · Authors · 2022-11-18
> **Response to Reviewer eZTL (1/2)**
>
> Thank you for the valuable reviews and the positive feedback on our paper! We respond to your questions and comments as follows:
>
> ---*"How FedDAR optimizes the learning objective in Equation 1 is unclear. Minimizing the expected risk averaged over all domains may not lead to a model which can perform well in every domain. This paper does not address fairness issues, nor does it achieve minimization of risks on a per-client basis, as each client has a data distribution that is a different mixture of predefined domains. While it is mentioned that the algorithm uses sample re-weighting, it is not clear how this is achieved with the current objective function."*
>
> As explained in Section 4.1, FedDAR optimizes the objective in Equation 1 by first decoupling the optimization for encoder and heads (from Equation 1 to Equation 5). The empirically estimation of risk (Equation 5) is achieved by re-weighting as we showed in Equation 6.
> Our learning objective emphasize the fairness between different domains since it puts equal weights on different domains. In contrast, a vanilla federated learning objective (i.e., FedAvg) implicitly weight each domain by the number of data points in the domain. Thus our objective is more fair domain-wisely. We are not aim on achieve minimization on a per-client basis since our goal is to minimize on a per-domain basis.
>
> ---*"The paper does not study the privacy implications of sharing model parameters."*
>
> We have added a paragraph to briefly discuss the privacy implications and the additional communication cost of our methods in the Appendix C. Following is what we added:
>
> **Communication** For FedDAR-WA, the only communication overhead comes from the extra parameters of multiple heads for different domains, which only slightly increase the communication cost. For FedDAR-SA, we need to send a Hessian matrix with $k^2\times N^2$ parameters from each client to the server at each round. This might be costly when both representation dimension $k$ and output dimension $N$ are large. However, compared to sending millions of parameters of neural network, the extra communication cost is acceptable.
>
> **Privacy** For the FedDAR-WA, there is no extra parameters shared compared to FedAvg. So there is no additional privacy risk introduced. Privacy techniques like homomorphic encryption [Cheon et al., 2017] or differential
> privacy [Kairouz et al., 2021] that apply to FedAvg also works for FedDAR-WA. In fact, the multi-head design of different domains makes it harder to perform gradient based attack[Zhu et al., 2019] targeting our method. Because the attacker need to first figure out which domain the sample comes from. For FedDAR-SA, the only extra parameters shared is the Hessian matrices, which are aggregated results from all the local data. Recovering the information for a specific sample from Hessian is extremely difficult. Under the worst circumstance, what the attacker can recover from the Hessian is the label and the features at last layer, which hardly ease the difficulty of recovering original input.
>
> [McMahan et al., 2017] McMahan, H. Brendan, et al. "Learning differentially private recurrent language models." arXiv preprint arXiv:1710.06963 (2017).
>
> [Kairouz et al., 2021] Kairouz, Peter, et al. "Practical and private (deep) learning without sampling or shuffling." International Conference on Machine Learning. PMLR, 2021.
>
> [Zhu et al., 2019] Zhu, Ligeng, Zhijian Liu, and Song Han. "Deep leakage from gradients." Advances in neural information processing systems 32 (2019).
>
> [Cheon et al., 2017] Cheon, Jung Hee, et al. "Homomorphic encryption for arithmetic of approximate numbers." International conference on the theory and application of cryptology and information security. Springer, Cham, 2017.

---

> > ### Comment · Reviewer_eZTL · 2022-12-01
> > **Thanks for the response**
> >
> > Thanks for the response. Below are my post-rebuttal thoughts.
> >
> > -The authors claim that they were able to reduce the sample complexity required by each domain to O(k) rather than O(k^2). While I have not checked the revised proof, their argument in the response is meaningful.
> >
> > -This paper does not directly maximize fairness between different domains. However, it tries to minimize the expected risk averaged over all domains, which emphasizes fairness by equally weighting risks of different domains. Another approach will be to choose a suitable fairness metric and try to maximize fairness. This paper puts a lot of emphasis on fairness. However, even after the response, the connection between fairness and optimizing domain performance and the trade-offs between these are somewhat unclear to me.
> >
> > -Communication and Privacy discussions are clear.

---

> ### Author Response · Authors · 2022-11-18
> **Response to Reviewer eZTL (2/2)**
>
> ---*”A lot of details regarding the convergence analysis are deferred to the appendix. It will be good to provide a sketch of the proof in the main paper. Please elaborate on techniques used in convergence analysis, highlighting the technical novelty of the results.“*
>
> Thanks for the suggestion. We agree it is helpful to include a sketch of the proof to facilitate the reader understanding our proof. However, we find it is very hard to squeeze the sketch into the main paper. So we defer it to the appendix. At a high level, we perform gradient descent for the $W^t$ term (domain-specific heads) and $\mathbf{B}^t$ term (embedding matrix) in an alternating manner; we note that the objective couples the two sequences together (since we need both $\mathbf{B}^t$ and $W^t$ to make a prediction). We then seek to show that the distance between the true embedding matrix $\mathbf{B}^*$ and our (gradient-descent based) estimate at time $t$, $\mathbf{B}^t$, satisfies an appropriate contraction inequality of the form $\mathrm{dist}(\mathbf{B}^*, \mathbf{B}^{t+1}) \leq \gamma \mathrm{dist}(\mathbf{B}^*, \mathbf{B}^t)$ for some $0 < \gamma < 1$ which depends on the initialization $\mathbf{B}^0$ as well as the problem parameters. In the process, we need to prove probabilistic bounds on various error terms (which arise due to access to only finitely many samples) so that the previously mentioned contraction inequality can hold; our final sample complexity (both per domain and overall) is determined largely by the number of samples required for these probabilistic bounds to hold (see e.g. the proof of Lemma A.11, and in particular the requirements on the sample size such that equations (26), (27) and (28) can hold).
>
> In addition, we note that we also tightened the analysis compared with [Collins et al., 2021], such that each domain only needs $O(k)$ samples for its domain-head update as opposed to $O(k^2)$ samples in [Collins et al., 2021] (in their paper which considers client heads rather than domain heads like us, they need each client-head update to have $O(k^2)$ samples). See our updated theoretical result in Theorem 4.1. This can yield a significant improvement when $k$ is moderately large and there is data imbalance.
>
> [Collins et al., 2021] Collins, Liam, et al. "Exploiting shared representations for personalized federated learning." International Conference on Machine Learning. PMLR, 2021.

---

### Official Review · Reviewer_XSkK · 2022-10-24

**Confidence:** 4
**Correctness:** 3
**Technical Novelty And Significance:** 4
**Empirical Novelty And Significance:** 3
**Recommendation:** 6

**Clarity, Quality, Novelty And Reproducibility:**

The paper is in general clear and well written, though it can use some proofreading to clean out some glaring typos. The novelty consisting on the domain-aware formulation and the theoretical results are satisfactory. Even though some of the aspects of the formulation are not properly justified the general concept is reasonable. The authors provide enough details in the main paper and supplementary material to make the experiments at least conceptually reproducible.

**Strength And Weaknesses:**

The theoretical results in Theorem 5.1 are interesting and highlight the sample requirements scaling with k^2, which is convenient, specially, if k << d as the authors argue. However, it is also important to recognize that modern applications involving (medical) images and text leverage encoders with representations whose dimensionality is in the high hundreds or lower thousands, which may be prohibitive for data-poor clients and in applications where sample size is limited, e.g., rare diseases and hard to obtain data such as pathology. Moreover, the large number of clients scenario described by the authors in which the cost decreases as n^-1 is not that realistic in healthcare domains.

The toy example in Section 3.1 is welcome, however, it is not necessarily clear or discussed why in the FedRep scenario, where heads are specified, one for each client, the representation is not flexible enough to account for the differences in domain. One can certainly devise a toy example where it is possible.

The need and motivation for u_m (inverse probability weighting) is not provided. Moreover, no intuition is provided about the difference or relationship between \hat{\cal R}_m and \hat{\cal R}_i, which only differ by u_m.

The justification for only using second order aggregation for the heads (with low enough dimensionality) is reasonable (accounting for communication costs), however, its relative benefit compared to only considering simpler aggregation for the encoder (which is understandable), is not provided, though explored in the experiments.

Though the use of the artificial data in Section 5.1 is justified, the considered generative process seems too simplistic to resemble any realistic scenario. Further, the results in Table 1 may be more impactful if the experiment is repeated multiple times to better understand the variation around the performance statistics (max, min and average accuracy). It will be also important to note the value of k used for the results in Table 1.

It is not necessarily clear whether AUC is a proper metric to assess fairness across domains. Perhaps the authors can elaborate on the reasoning behind their fairness claim.

The results in Table 3 are impressive, however, it is not clear why the authors used a cross-validation scheme considering that one of the claims is that the proposed approach is not data hungry. Further, the dimensionality of the EXAM data is not specified.

**Summary Of The Paper:**

The authors address the problem of personalized (to a client) federated learning in which the distribution from each client is assumed to be drawn from a mixture of predefined domains. The proposed model learns a shared data representation with independent prediction modules (heads) for each client. Moreover, the authors show that in the linear case, the proposed approach enjoyed linear convergence and empirically, that on both artificial and real-world data, the proposed approach outperforms existing federated learning methods.

**Summary Of The Review:**

The authors presented a domain-aware approach for federated learning together with a theoretical analysis of the (per iteration) sample complexity of the methodology and empirical results using both synthetic (sections 5.1 and 5.2) and convincing real-world data (Section 5.3). The contributions of each of the components of the model are justified with an ablation study in Section 5.3. Though the theoretical results are interesting, they may not be applicable to many real-world scenarios; and though the weights u_m seem reasonable, they are not justified or motivated.

---

> ### Author Response · Authors · 2022-11-18
> **Response to Reviewer XSkK (1/3)**
>
> We sincerely thank the reviewer for the critical feedback! We address the concerns raised by the reviewer as follows:
>
> ---*"The theoretical results in Theorem 5.1 are interesting and highlight the sample requirements scaling with $k^2$, which is convenient, specially, if $k << d$ as the authors argue. However, it is also important to recognize that modern applications involving (medical) images and text leverage encoders with representations whose dimensionality is in the high hundreds or lower thousands, which may be prohibitive for data-poor clients and in applications where sample size is limited, e.g., rare diseases and hard to obtain data such as pathology. Moreover, the large number of clients scenario described by the authors in which the cost decreases as $n^{-1}$ is not that realistic in healthcare domains."*
>
> We are glad that the reviewer find our theoretical result interesting, and thank the reviewer for this thoughtful question. After a careful review of our theoretical results, we found that the sample complexity required by each domain can actually be reduced to be of order $O(k)$ rather than of order $O(k^2)$; this is reflected in Lemma A.11 in the Appendix, where we note that eq. (24) corresponds to the updated lower bound required for the number of samples per domain. The bound in eq. (24) scales with $k$ rather than $k^2$. At a technical level, the main change that enables the improvement is a tightening of the probabilistic argument in bounding the error quantity $\|F_m\|$ in Appendix A.4.2. We have updated our results in the revision illustrating the improvement in sample complexity; the changes (which only take place in a few places) are primarily in Appendix 4.2, and all made in blue. We note that the overall sample complexity (per client) remains $\tilde{O}(dk^2/n)$. The reduction in per-domain sample complexity to $\tilde{O}(k)$ implies that our results can better handle domain imbalance (e.g. for rare diseases with less data) when $k$ is in the high hundreds or lower thousands.
>
> As the reviewer correctly points out, in healthcare applications,  $n$ (number of clients) may indeed not be large, since each client may correspond to a large entity, e.g. a healthcare. However, in such cases, since each client is a large entity, it is likely that a single client may provide many samples. Nonetheless, we acknowledge that in data sparse environments, if the overall number of samples is less than $O(dk^2)$, our theorem cannot guarantee FedDAR's convergence. However, empirically, we have found that our method does not require that many samples to work even when $k$ is large (such that the overall number of samples required, $O(dk^2)$, is large). For instance, in our experiments with FedDAR-WA, the dimensionality of representations are $512$ and $278$ for the models we used in FairFace and EXAM datasets respectively. In addition, further savings on the number of samples required can be obtained by considering $k$ as a hyper-parameter to be tuned, by simply adding one fully-connected layer on top of the original representation. Indeed, we showed that using a properly tuned small $k$ can improve the performance in Figure 2 and Table 2.
>
> ---*"The toy example in Section 3.1 is welcome, however, it is not necessarily clear or discussed why in the FedRep scenario, where heads are specified, one for each client, the representation is not flexible enough to account for the differences in domain. One can certainly devise a toy example where it is possible."*
>
> We note that our domain mixture assumption specifies that the shift between clients' distributions are caused by the different mixture coefficients of domains. Further, as mentioned in our paper (Section 3), we assume each domain has a different conditioned label distribution, i.e., $P_m(y|x)$ is different in each domain $m$. Under this setting, one cannot rely on a flexible representation to account for the difference in domain and expect a single classifier on top of the representation can handle all domains.
>
> Our toy example in Section 3.1 articulates a scenario where a linear model is used and FedRep is sub-optimal. In original paper of FedRep [Collins et al., 2021], a similar linear model toy example is employed to demonstrate the advantage of FedRep over FedAvg.
>
> [Collins et al., 2021] Collins, Liam, et al. "Exploiting shared representations for personalized federated learning." International Conference on Machine Learning. PMLR, 2021.

---

> ### Author Response · Authors · 2022-11-18
> **Response to Reviewer XSkK (2/3)**
>
> ---*"The need and motivation for $u_m$ (inverse probability weighting) is not provided. Moreover, no intuition is provided about the difference or relationship between $\hat{\cal R}_m$ and $\hat{\cal R}_i$, which only differ by $u_m$."*
>
> $\hat{\cal{R}}_m$ and $\hat{\cal {R}}_i$ are the empirical risk for domains and clients respectively. We want our model to minimize the expected risk averaged over all domains $\frac{1}{M}$ $\sum_m \hat{\cal R}_m$ instead of all clients. The motivation of doing this is to emphasize the fairness between different domains since it puts equal weights on different domains. In contrast, a vanilla federated learning objective (i.e., FedAvg: $\sum_i\frac{L_i}{L}\hat{\cal R}_i$) implicitly weight each domain by the number of data points in the domain. In the real-world applications from fields like healthcare, some of the data domain can be highly underrepresented, e.g. the minority ethic groups in the population. Therefore, our objective is more fair domain-wisely. We demonstrate this empirically in the updated Table 1 and Table 2 where adding the reweighting brings considerable improvement for the domain with the lowest accurracy/AUC.
>
> ---*"The justification for only using second order aggregation for the heads (with low enough dimensionality) is reasonable (accounting for communication costs), however, its relative benefit relative to only considering simpler aggregation for the encoder (which is understandable), is not provided, though explored in the experiments."*
>
> The idea of second order aggregation is inspired by the Newton method for accelerating optimization. Although not rigorously proved, conceptually, second order aggregation leverage the local geometry information (Hessian matrix) thus can converge faster than the simpler aggregation.
>
> ---*"Though the use of the artificial data in Section 5.1 is justified, the considered generative process seems too simplistic to resemble any realistic scenario."*
>
> The artificial data in Section 5.1 is oversimplified for theoretical purpose, and servers only as a test bed for our theoretical results. We later use real data with controlled distribution and real distribution to demonstrate our method in realistic scenarios.
>
> ---*"Further, the results in Table 1 may be more impactful if the experiment is repeated multiple times to better understand the variation around the performance statistics (max, min and average accuracy). It will be also important to note the value of k used for the results in Table 1."*
>
> Thanks for the suggestion. The results in Table 1 is now reported with mean and standard error from three independent runs with different random seeds. Thanks for catching the missed value of $k$ in Table 1. It is 8. We fixed it in our revision.
>
> ---*"It is not necessarily clear whether AUC is a proper metric to assess fairness across domains. Perhaps the authors can elaborate on the reasoning behind their fairness claim."*
>
> For the the fairness claim, in fact, there is no standard metric to assess fairness across domains in healthcare field yet. People usually compares some performance metrics like true positive rates (TPR), area under the curve (AUC) metric, etc across domains to assess the fairness[Feng et al., 2022]. The metric to use actually depends on the specific task and the goal you want to achieve with your model. We use AUC since it is the major metric used in the original EXAM paper[Dayan et al., 2021]. It is widely used in healthcare field and a well-suited metric for the specific task with EXAM dataset. The task is binary classification with imbalanced label distribution (around 10\% positive cases), in this case, accuracy is not a good choice. For the Fairface we used accuracy since the label distribution is more balanced.
>
> [Feng et al., 2022] Feng, Qizhang, et al. "Fair Machine Learning in Healthcare: A Review." arXiv preprint arXiv:2206.14397 (2022).
>
> [Dayan et al., 2021] Dayan, Ittai, et al. "Federated learning for predicting clinical outcomes in patients with COVID-19." Nature medicine 27.10 (2021): 1735-1743.

---

> ### Author Response · Authors · 2022-11-18
> **Response to Reviewer XSkK (3/3)**
>
> ---*"The results in Table 3 are impressive, however, it is not clear why the authors used a cross-validation scheme considering that one of the claims is that the proposed approach is not data hungry. Further, the dimensionality of the EXAM data is not specified."*
>
> The major reason we report results on 5-fold cross validation is the limited number of positive cases, especially for those under-represented ethic groups. From the experiments we observe that the performance of some baselines are very sensitive to different data splits. We want to make sure that the  variation coming from different data split is shown in the results. Thanks for noticing the missing of specification of the EXAM data. The input is one chest x-ray image resized to 224x224 paired with a 22-dimensional electronic health record(EHR) data, the representation dimension is $278$ if it is not projected. We have added this in our revision.

---

### Official Review · Reviewer_GcPS · 2022-10-25

**Confidence:** 3
**Correctness:** 3
**Technical Novelty And Significance:** 2
**Empirical Novelty And Significance:** 2
**Recommendation:** 6

**Clarity, Quality, Novelty And Reproducibility:**

quality
The proposed method is well analyzed and discussed, but there are insufficient comparisons and experiments to support the claims.

clarity
The paper is well organized.

originality
The work is less original as there are numerous works that share similar ideas with the proposed method.


**Strength And Weaknesses:**

Strength
1. solid theoretical analysis on convergence.

2. extensive experiments on both synthetic data and real-world datasets validate the effectiveness of the proposed method.

Weaknesses
1. The domain shift problem in FL has been extensively discussed in various works in recent years, such as [1-3]. Specially, [4] proposed a similar architecture where a representation module is shared and multiple classifiers heads are applied for target domains. As these methods are not mentioned or compared, it is uncertain if the contributions of the proposed FedDAR hold and the experiments are not convincing.

2. This work assumes that there is labeled information about the domain a sample belongs to, which limits the proposed FedDAR’s application as it is usually not the case in FL. Besides, as there are no comparisons with those unsupervised methods[1,3] handling the domain shift problem in FL, it is unclear how many improvements the labeled information brings.

3. As discussed in Sec.1 and Sec 2., personalized models have been recognized to have superior performance to vanilla FL models. Then comparisons in Sec.3.1, Sec.5.1, and Sec.5.2 would be less persuasive as they compare the FedDAR with vanilla FedAvg rather than other personalized methods. For example, will FedDAR outperform a vanilla model fine-tuned on a client’s local data?

4. The second-order aggregation shares similar ideas as those personalization methods with a meta-learner [5]. Besides, as sharing and updating a Hessian matrix would be communication expensive and unstable, it requires a client to have sufficient training data and limits FedDAR to have simple classifier heads. Experiments in Appendix B.2. also indicates this problem.

[1] Disentangled Federated Learning for Tackling Attributes Skew via Invariant Aggregation and Diversity Transferring (https://arxiv.org/abs/2206.06818)

[2] FedBN: Federated Learning on Non-IID Features via Local Batch Normalization (https://arxiv.org/abs/2102.07623)

[3] Federated Adversarial Domain Adaptation (https://arxiv.org/abs/1911.02054)

[4] Diurnal or Nocturnal? Federated Learning of Multi-branch Networks from Periodically Shifting Distributions (https://openreview.net/forum?id=E4EE_ohFGz)

[5] Personalized Federated Learning: A Meta-Learning Approach (https://arxiv.org/abs/2002.07948)



**Summary Of The Paper:**

In this paper, the authors propose a domain-aware representation learning method (FedDAR) for the non-iid FL problem. The FedDAR assumes data on clients are from multiple domains and learns a classifier head for each domain. A representation module is shared for all classifier heads and updated by the vanilla FedAvg. The authors also proposed a second-order aggregation method to update domain-aware classifier heads, whose effectiveness is validated by ablation studies. Experiments on both synthetic data and real-world datasets validate the effectiveness of the proposed method.







**Summary Of The Review:**

This paper proposes a domain-aware representation learning method (FedDAR) for the non-iid FL problem. The authors provide solid convergence analysis and validate FedDAR’s performance on both synthetic and real-world datasets. However, assumptions on domain labels and simple classifier heads limit the application of FedDAR. Besides, as there are limited comparisons with existing methods sharing similar ideas or settings, the paper’s contributions could be limited, and the results are less persuasive.

---

> ### Author Response · Authors · 2022-11-18
> **Response to Reviewer GcPS (1/2)**
>
> We thank the reviewer for the comments. We are glad to see that the reviewer finds our paper has "solid theoretical analysis on convergence" and "extensive experiments on both synthetic data and real-world datasets validate the effectiveness of the proposed method".
>
> The major concern of the reviewer is originality. The reviewer thinks there are numerous works that share similar ideas with our work, and provides us with 5 prior works. However, we would like to clarify that these papers do not share the key idea of our work. The reviewer may misunderstand the relationship between our work and those prior works due to the similar keywords used in these papers, which actually has very different meanings.
> %by mistakenly considering two works sharing similar keywords also share similar ideas.
>
> In the following, we explain the difference between our work and the 5 mentioned prior works. First, our problem is different. We consider a specific non-iid federated learning (FL) setting where each client's distribution is a mixture of predefined domains. Such a specific assumption is the core of our work and has never been studied by prior works including [1-5].
>
> Further, [3] considers a totally different problem which is domain adaptation in a federated learning manner. Thus it is not in the scope of our paper.
> [1,2] studies federated learning under the assumption of FedRep [6] while proposing different techniques to solve it: [1] uses disentangled feature, [2] uses local batch normalization. As we discussed in Sec 3.1, FedRep has intrinsic disadvantages in our specific setting.
>
> [4] is aimed to handle a different specific FL setting where the clients can be clustered into several groups that share the same data pattern. [4] uses a multi-branch network to handle each cluster of clients and each client will be handled by the one branch corresponding to its group. We concede that the idea of using a multi-head network for model personalization is similar. But the domain distribution assumptions are totally different in [4] and our work. In [4], one client can only have data from on domain, while in our setting, one client has data from multiple domains. Thus each client will use all the branches of our model to handle data from different domains.
>
> In Section 2 related work, we did mentioned [2]. We added [1] and [4] to Section 2 in our revision, since they both fall into the category of client-wise personalized FL.
>
> The reviewer claim that our idea of second-order aggregation is similar to [5] because both works use Hessian matrix in the algorithms. This is not the case. First, the optimization problem of [5] (Equation 3 in [5]) is different from ours (Equation 1 in our paper). Second, the two works have different motivations to use the second-order information. [5] uses Hessian matrix since meta-learning requires second-order operation to optimize the meta parameters. While we introduce second-order operation, inspired by the Newton method, to speed up the learning process of head classifiers. Third the two algorithms use the Hessian matrix in different ways. The algorithm in [5] uses the Hessian to locally updates the model parameters and uses simple average to aggregate local models. In contrast, our FedDAR compute the Hessian matrices after the locally updates are done, and send all the local Hessian matrices to the server for aggregating. Hence there is no similarity between our ideas and [5].
>
> In summary, our work is very different from the prior works. We believe our work provides a novel perspective of non-iid federated learning (FL) setting where each client's distribution is a mixture of predefined domains. While all prior personalized FL methods, including [1,2,4], assume that data from one client can only come from one domain.
>
> [1] Luo, Zhengquan, et al. "Disentangled Federated Learning for Tackling Attributes Skew via Invariant Aggregation and Diversity Transferring." arXiv preprint arXiv:2206.06818 (2022).
>
> [2] Li, Xiaoxiao, et al. "Fedbn: Federated learning on non-iid features via local batch normalization." arXiv preprint arXiv:2102.07623 (2021).
>
> [3] Peng, Xingchao, et al. "Federated adversarial domain adaptation." arXiv preprint arXiv:1911.02054 (2019).
> [4] Zhu, Chen, et al. "Diurnal or Nocturnal? Federated Learning of Multi-branch Networks from Periodically Shifting Distributions." International Conference on Learning Representations. 2021.
>
> [5] Fallah, Alireza, Aryan Mokhtari, and Asuman Ozdaglar. "Personalized federated learning: A meta-learning approach." arXiv preprint arXiv:2002.07948 (2020).
>
> [6] Collins, Liam, et al. "Exploiting shared representations for personalized federated learning." International Conference on Machine Learning. PMLR, 2021.

---

> ### Author Response · Authors · 2022-11-18
> **Response to Reviewer GcPS (2/2)**
>
> We address the other comments from the reviwer as follows:
>
> ---*"This work assumes that there is labeled information about the domain a sample belongs to, which limits the proposed FedDAR’s application as it is usually not the case in FL. Besides, as there are no comparisons with those unsupervised methods[1,3] handling the domain shift problem in FL, it is unclear how many improvements the labeled information brings."*
>
> As mentioned above, [3] considers a totally different problem, thus cannot be used as a baseline. For [1], although it clusters clients into different domains in a unsupervised fashion, it still assumes that data from one client can only come from one domain. We could included it as a baseline like other prior client-wise personalized method, but unfortunately [1] is a concurrent work and not open-source, it is difficult for us to do so. We add FedBN [2] to our baselines in our revision. From the comparison, we can see that our FedDAR outperforms FedBN by a clear margin. As shown in Table 3, FedDAR has about $5\%$ to $10\%$ higher AUCs for different domains in the EXAM dataset.
>
> ---*"comparisons in Sec.3.1, Sec.5.1, and Sec.5.2 would be less persuasive as they compare the FedDAR with vanilla FedAvg rather than other personalized methods."*
>
> We already compared our method with a personalized method FedRep in Sec3.1 and Sec 5.1. The result for FedRep in Sec3.1 can  be extended to works like [1] and [2] because they share similar objective. We didn't include more personalized methods as baselines in Sec 5.2 because the major purpose of it is to understand the behavior of our method under different level of data heterogeneity with controlled domain distribution. We think it is more persuasive to demonstrate the superiority over other personalized methods on real-world data with real data distribution instead of simulated data distribution. Therefore, the comparison with other personalized methods is only made in Sec 5.3.

---

### Author Response · Authors · 2022-11-18
**Summary of Updates**

We thank all the valuable comments from the reviewers. We have revised our paper accordingly. All the updates are in blue color for a clear view. We summarize the main updates as follows:
1. Fixing typos and notations in Algorithms 1.
1. Updating our theoretical results. The per-domain sample complexity is now reduced to $\tilde{O}(k)$.
1. Adding proof sketch in Appendix A.3.
1. Adding missing experimental details: i) $k$ used in Table 1. ii) the dimensionality of the EXAM data. iii) how hyper-parameters are tuned.
1. Figure 1 in Sec 5.1 is updated with error bars and results using separate Fedavg for each domain.
1. Figure 2 in Sec 5.2 is updated with error bars.
1. Adding results of Fedavg without domain-reweighting, separate Fedavg in Table 1. Adding standard errors from three independent runs to results.
1. Adding FedBN as a baseline in Table 3.
1. Adding a section discussing the privacy implication and communication cost in Appendix C.

---

### Decision · Program_Chairs · 2023-01-20

**Decision:**

Accept: poster

**Justification For Why Not Higher Score:**

 it's well-deserved acceptance, but it didn't show any more surprising results than that.

**Justification For Why Not Lower Score:**

 n/a

**Metareview: Summary, Strengths And Weaknesses:**

The paper considers a specific non-iid federated learning environment where each client's distribution is a mixture of predefined domains. To prevent the performance drop of the FL method in this environment, the authors propose a method called FedDAR consisting of a shared encoder across all domains and a personalized head in each domain.

All reviewers, including this AC, somewhat agreed that the setting is important, idea is novel and the paper is well written in most places, and unanimously suggested accept. However, even through rebuttal, reviewers were not completely convinced on some issues such as fairness metric/connection to optimization objective. It would be nice if these concerns could be completely resolved in the final version. And also in the final version, I think a concept figure about your method is absolutely necessary to help readers understand.

**Note From Pc:**

if the above contains the word "oral" or "spotlight" please see: "oral" presentation means -> notable-top-5% and "spotlight" means -> notable-top-25%. As stated in our emails, we are disassociating presentation type from AC recommendations